# Structures of the ApoL1 and ApoL2 N-terminal domains reveal a non-classical four-helix bundle motif

Mark Ultsch[1], Michael J. Holliday [2], Stefan Gerhardy[2], Paul Moran [2], Suzie J. Scales [3], Nidhi Gupta [3], Francesca Oltrabella [3], Cecilia Chiu [4], Wayne Fairbrother [2], Charles Eigenbrot [1] & Daniel Kirchhofer [2✉]

Apolipoprotein L1 (ApoL1) is a circulating innate immunity protein protecting against trypanosome infection. However, two ApoL1 coding variants are associated with a highly increased risk of chronic kidney disease. Here we present X-ray and NMR structures of the N-terminal domain (NTD) of ApoL1 and of its closest relative ApoL2. In both proteins, four of the five NTD helices form a four-helix core structure which is different from the classical four-helix bundle and from the pore-forming domain of colicin A. The reactivity with a conformation-specific antibody and structural models predict that this four-helix motif is also present in the NTDs of ApoL3 and ApoL4, suggesting related functions within the small ApoL family. The long helix 5 of ApoL1 is conformationally flexible and contains the BH3-like region. This BH3-like α-helix resembles true BH3 domains only in sequence and structure but not in function, since it does not bind to the pro-survival members of the Bcl-2 family, suggesting a Bcl-2-independent role in cytotoxicity. These findings should expedite a more comprehensive structural and functional understanding of the ApoL immune protein family.

[1] Department of Structural Biology, Genentech Inc., South San Francisco, CA, USA. [2] Department of Early Discovery Biochemistry, Genentech Inc., South San Francisco, CA, USA. [3] Department of Immunology, Genentech Inc., South San Francisco, CA, USA. [4] Department of Antibody Engineering, Genentech Inc., South San Francisco, CA, USA. ✉email: dak@gene.com

Apolipoprotein 1 (ApoL1) is an innate immunity protein belonging to the ApoL family[1–3], which comprises six members. The expression of ApoL family proteins is strongly induced by cytokines, such as interferons and tumor necrosis factor[4–6] and, therefore, the entire ApoL family may function in innate immune defense[3,6–8]. ApoL1, which is present only in humans and some nonhuman primates[9–12], is the only secreted family member, circulating in blood associated with high density lipoprotein (HDL) particles, or as an IgM complex and protects against infection of the trypanosome subspecies *T. brucei brucei*[13–15]. The two allelic variants ApoL1-G1 and ApoL1-G2, additionally protect against sleeping sickness caused by *T. brucei gambiense*[16] and *T. brucei rhodesiense*[17], respectively. Unlike the ApoL1-G0, the G2 variant no longer binds to the trypanosome serum resistance-associated protein (SRA)[18,19] and, thus, avoids neutralization by trypanosome SRA[18,20,21]. On the other hand, homozygous carriers of these allelic variants have a heightened risk for developing chronic kidney disease[18,22]. Podocyte-specific overexpression of these ApoL1 risk variants in mice was shown to mimic the human disease[23]. However, there is no accord on the disease mechanism. Diverse subcellular locations and cytotoxic pathways were proposed, including autophagy[23–26] and apoptosis[27] (reviewed by refs.[28–35]). At the molecular level, ApoL1 was shown to form ion channels requiring acidic-pH-driven membrane insertion followed by channel activation at neutral pH[36–44]. The cytotoxic activity of the risk variants is initiated by the influx of calcium and sodium ions by plasma membrane-embedded ApoL1-G1 and -G2 channels, resulting in the activation of cell death pathways[42]. However, the molecular details of channel formation, the channel structure, composition, and its regulation remain largely unknown.

According to secondary structure predictions, ApoL1 is composed of amphipathic α-helices[1,2] and contains three or four transmembrane domains[19,43]. The C-terminal region, known as the SRA interacting domain (SRA-ID), contains a leucine zipper motif and interacts with the monomeric trypanosome surface glycoprotein SRA[45,46], but also with ApoL3[47,48] and the vesicle-associated membrane protein 8[49]. The extended N-terminal region (also known as the pore-forming domain (PFD)[39]) encompasses a putative transmembrane segment[19,43] and was predicted to adopt a colicin A-like fold[39]. This led to the proposition that the ApoL1 pore-forming mechanism is related to that of pore-forming colicins, diphtheria toxin, and B-cell lymphoma 2 (Bcl-2) family members[8,39], which share structural similarities[50]. However, this long-standing model may need revision in light of a recent study by Schaub et al.[43], which provides strong evidence that the pore-lining region is actually located in the C-terminal, rather than the N-terminal region of ApoL1. A Bcl-2 homology domain 3 (BH3)-like motif[8,26] located upstream of the helix-loop-helix (H-L-H) transmembrane domain region (residues 177–228) was predicted for ApoL1 and for other ApoL family members[8,51,52], including all murine homologs[7]. The canonical BH3-only proteins[53], such as Bim and Bid, are ligands of the Bcl-2 family of pro-survival proteins[54,55] regulating apoptosis and autophagic cell death[54]. Similarly, the BH3-like motifs of ApoL1, ApoL6, and some of the murine ApoL homologs were found to play a role in apoptotic and autophagic cell death[7,25,26,52]. In contrast, the BH3-like region of ApoL2 did not induce cell death or autophagy[51] and other studies concluded that the BH3-like motif is not required for ApoL1-mediated cytotoxicity[56,57].

The structural knowledge of ApoL1 is quite limited and is mainly based on computational models of different ApoL1 regions[17,39,49,58,59]. In addition, no inference can be made from other ApoL family members, since their structures are also unknown. The SRA-ID has attracted attention since it harbors the mutations associated with kidney disease: the ApoL1-G1

mutation S342G:I384M and the ApoL-G2 deletion of residues N388 and Y389 at the very C-terminus. An NMR study of the SRA-ID demonstrated that it is well-structured in solution[58] and structural models predicted a coiled coil conformation[17,49,58]. However, the studies are at variance with respect to the structural influences of the disease-causing mutations G1 and G2, which were found to either stabilize[49] or destabilize[58] this conformation, perhaps due to the different lengths of SRA-ID constructs used by these groups[60]. Notwithstanding, the coiled coil model seems in conflict with the recently proposed cation channel model in which the transmembrane pore-lining region (L335–S356)[43], roughly corresponding to the N-terminal helix of the coiled coil model, would be sequestered within the membrane and, thus, unable to participate in a coiled coil structure.

Therefore, to obtain a better structural understanding of ApoL1 and other ApoL family members, we investigated the structures of the N-terminal domain (NTD) of ApoL1 and ApoL2 by crystallographic and NMR studies. In ApoL1, this domain comprises the soluble portion of the N-terminal region (D61–T172) ending a few residues before the two predicted transmembrane domains, the H-L-H region[19,43]. The five Fab co-crystal structures and a solution NMR structure described herein provide a comprehensive view of this region. The NTDs of both ApoL1 and ApoL2 adopt the same fold composed of four amphipathic helices, which is a conformation unique to members of the ApoL family and clearly distinct from the previously reported colicin fold. The herein reported structural and bio-physical findings revise previous computational models and conceptions of this ApoL1 region.

## Results

### Antibodies as chaperones for crystallization of the ApoL1-NTD.
Initial crystallization attempts were carried out with full-length ApoL1 and with two truncated forms encompassing the H-L-H region, all of which were only soluble in the presence of detergents. Extensive crystallization trials were carried out with these ApoL1 constructs as apo-forms or in complex with antibody Fabs and in the presence of various detergents and lipids. However, these efforts were unsuccessful, as we were unable to obtain diffracting crystals (see "Methods" for details). Subsequently, we focused our attention on an N-terminal construct (D61–T172), which was soluble in the absence of detergents. The D61 residue was selected as the N-terminal boundary because the preceding residues (E28–M60) were not predicted to have secondary structure. This construct, which is herein referred to as the ApoL1 N-terminal domain (ApoL1-NTD), encompasses most of the so-called PFD[39], except the predicted transmembrane H-L-H segment (residues 177–228)[19,43]. The protein, which was purified from insect cells, eluted as a monomer of 15 kDa by size-exclusion chromatography coupled to multi-angle laser light scattering (SEC-MALS) (Supplementary Fig. 1a). Crystallization trials of ApoL1-NTD were not successful and, therefore, we chose three antibodies 3.6D12 (Ab6D12), 3.3B6 (Ab3B6), and 3.7D6 (Ab7D6)[61] as crystallization chaperones. These antibodies were previously identified to bind to the ApoL1-NTD region[61] and recognize both ApoL1 associated with HDL particles and ApoL1 on the podocyte cell surface[61]. All three antibodies showed strong binding to ApoL1 by surface plasmon resonance (SPR), having $K_D$ values of 0.86, 1.56, and 1.63 nM, respectively (Supplementary Table 1).

### Co-crystal structures of ApoL1-NTD reveal a four-helix core conformation and the C-terminal BH3-like helix
*A. The Fab6D12:ApoL1-NTD complex.* Crystals of the Fab6D12:ApoL1-NTD complex grew in space group P4₃ with two complexes per asymmetric unit (ApoL1 chains M and K; rmsd of

**Table 1 Data collection and refinement statistics of crystal structures.**

| | Fab6D12:ApoL1-NTD | Fab3B6:ApoL1-NTD | Fab7D6:ApoL1-NTD | Fab7D6:ApoL1-peptide | Fab6D12:ApoL2-NTD |
|---|---|---|---|---|---|
| **Data collection** | | | | | |
| Space group | P4$_3$ | P1 | C2 | P 3$_1$ 2 1 | C 2 2 2$_1$ |
| Cell dimensions | | | | | |
| $a, b, c$ (Å) | 131.10 131.10 87.16 | 52.75 71.87 123.44 | 94.24 60.26 124.22 | 129.95 129.95 90.82 | 97.90 154.29 88.05 |
| $\alpha, \beta, \gamma$, (°) | 90, 90, 90 | 80.16, 85.05, 90.16 | 90, 111.82, 90 | 90, 90, 120 | 90, 90, 90 |
| Resolution (Å) | 41.46–2.03 (ellipsoidal) 41.46–2.26 (isotropic) (2.152–2.026)$^a$ | 70.8–1.857 (ellipsoidal) 70.53–2.42 (isotropic) (1.924–1.857)$^a$ | 49.63–1.91 (1.98–1.91)$^a$ | 70.67–2.16 (ellipsoidal) 47.81–2.38 (isotropic) (2.234–2.157)$^a$ | 32.73–2.15 (ellipsoidal) 48.7–2.87 (isotropic) (2.225–2.148)$^a$ |
| $R_{merge}$ | 0.087 (2.297) | 0.059 (0.496) | 0.060 (0.370) | 0.149 (1.638) | 0.286 (1.776) |
| $I/\sigma I$ | 12.8 (1.5) | 3.58 (1.6) | 9.7 (2.5) | 11.3 (1.2) | 17.7 (1.4) |
| Completeness (%) | 95.1 (64.4) (anisotropic) | 89.6 (55.2) (anisotropic) | 98.6 (99.9) | 89.6 (43.5) (anisotropic) | 93.5 (69.0) (anisotropic) |
| Redundancy | 4.9 (4.5) | 3.3 (3.3) | 3.3 (3.2) | 9.4 (7.8) | 8.5 (6.3) |
| **Refinement** | | | | | |
| Resolution (Å) | 41.46–2.03 (anisotropic) | 70.8–1.857 (anisotropic) | 49.63–1.91 | 70.67–2.16 (anisotropic) | 32.73–2.15 (anisotropic) |
| No. reflections | 76,896 (3846) | 82,729 (577) | 494,667 (7276) | 40,649 (1294) | 25037 (1254) |
| $R_{work}/R_{free}$ | 0.1886/0.2172 | 0.2047/0.2416 | 0.1677/0.1970 | 0.1655/0.2028 | 0.1920/0.2300 |
| No. atoms | | | | | |
| Protein | 7995 | 7851 | 4034 | 3467 | 3749 |
| Ligand/ion | 30 | 14 | 34 | 35 | 11 |
| Water | 345 | 386 | 406 | 185 | 71 |
| *B*-factors | | | | | |
| Protein | 62.80 | 52.50 | 42.6 | 45.80 | 41.70 |
| Ligand/ion | 81.50 | 37.50 | 61.2 | 70.70 | 89.40 |
| Water | 49.60 | 35.80 | 43.4 | 49.60 | 33.40 |
| R.m.s. deviations | | | | | |
| Bond lengths (Å) | 0.004 | 0.011 | 0.006 | 0.006 | 0.014 |
| Bond angles (°) | 0.63 | 1.37 | 5.35 | 4.04 | 0.13 |

One crystal was used for each structure.
$^a$Values in parentheses are for highest-resolution shell.

0.5 Å for 1080 atoms) and diffracted to 2.0 Å (data collection and refinement statistics in Table 1). No electron density was observed for the four N-terminal residues D61–S64 or the C-terminal segment K142–T172. The resolved portion of ApoL1-NTD spanning residues S65–L141 is composed of four α-helices connected by three short turns of 1–4 amino acids (Fig. 1a). The four helices that form the core structure (S65–I123) are amphipathic with the hydrophobic residues pointing toward the interior and engaging in hydrophobic interactions (Fig. 1b). They form a loosely packed bundle in which the individual helices diverge from each other, rather than being aligned in a parallel lengthwise orientation as found in the classical four-helix bundle[62]. The C-terminal half of the long helix 4 (M124–L141) consists of mostly charged residues and packs against a symmetry-related molecule (chain K′) in the crystal lattice (Fig. 1a) and, therefore, this region of helix 4 may have adopted a conformation influenced by these contacts and may not represent an important natural state.

The Fab6D12-binding epitope (V98–D109) is compact and relatively small, with ApoL1-NTD having a buried solvent-accessible surface of 540 Å$^2$. Only a few residues engage in Fab6D12 interactions and are located on helices 3 and 4 and the loop connecting them: A99, V98, R105 (hydrophobic), and L103 and N106 (H-bonds) (Fig. 1b, c; electron density in Supplementary Fig. 2a). The most important interaction is made by the positively charged R105, which inserts into a deep acidic pocket formed by CDR-H3 and CDR-L1 loops (Fig. 1d). The aromatic rings of HC-Y103 and LC-Y32 are arranged in parallel and sandwich the aliphatic side chain portion of R105 (Fig. 1c). An ApoL1-NTD mutant in which the R105 was changed to an alanine residue showed a strong binding loss in a biolayer

interferometry experiment, confirming a key role of R105 in Fab6D12 binding (Supplementary Fig. 1b).

**B. The Fab3B6:ApoL1-NTD complex.** Crystals of the Fab3B6:ApoL1-NTD complex grew in space group P1 with two complexes per asymmetric unit and diffracted to 1.86 Å (data collection and refinement statistics in Table 1). Electron density for ApoL1 chain A was observed from NTD residue I66 to R159 and for ApoL1 chain C from E69 to V168 (Fig. 2a). The difference between the two chains can be attributed to crystal packing, with the C-terminus of chain A being exposed to solvent, whereas the C-terminus of chain C is engaged in additional contacts with a symmetry-related Fab. Chain C also includes residues K142–V168, not seen in the Fab6D12 complex structure. This segment is part of the long helix 5 that has a kink due to the helix-breaking Pro145 and ends with the BH3-like region (N154–Q166) (Fig. 2a). A superposition with the Fab6D12:ApoL1-NTD structure shows that the four-helix core (I66–I123) is fully preserved (rmsd of 0.6 Å for 540 atoms) including the Fab6D12 epitope region (V98–D109) (Fig. 2a). The main difference between the two structures is that the long helix 4 of the Fab6D12 structure is bent in the Fab3B6 structure and that this new "elbow" harbors the Fab3B6-binding epitope (I123–R137) (yellow in Fig. 2a). The epitope region comprises four basic residues (K125, K127, K132, and R137) (electron density in Supplementary Fig. 2b) that engage in salt bridge or H-bond interactions, as well as hydrophobic interactions via their aliphatic portions (K125 and K127). The paratope surface is predominantly acidic, due to four acidic residues L2-D54, H2-D55, H2-D57, and H3-E99, which all engage in interactions with the

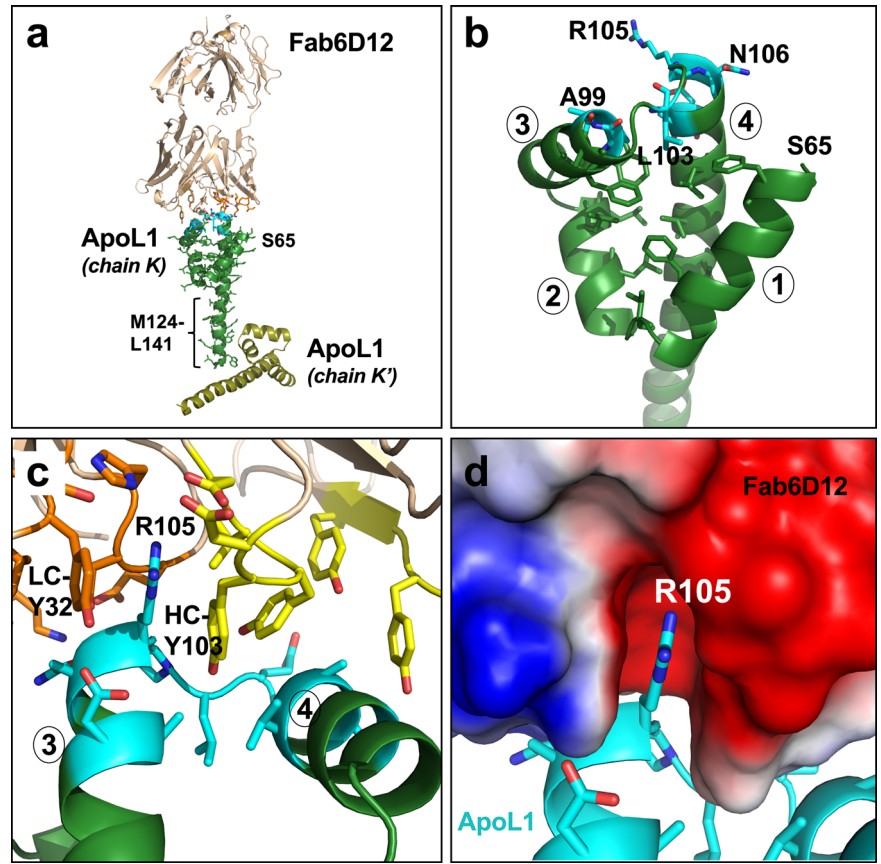

**Fig. 1 Structure of the Fab6D12:ApoL1-NTD complex. a** Overview of the complex with Fab6D12 in brown with contact residues (≤4-Å distance) as orange sticks and ApoL1-NTD in dark green (chain K) with contact residues as cyan sticks. The first resolved N-terminal residue of the ApoL1-NTD (D61–T172) is S65. Shown in yellow-green is a symmetry-related ApoL1-NTD molecule (chain K') in the crystal lattice, which stabilizes the C-terminal helical segment (M124–L141). **b** Close-up view of the four-helix core (helices 1–4 are indicated) with key epitope residues as cyan sticks. The buried hydrophobic residues of the four amphipathic helices, which stabilize the four-helix conformation are shown as dark green sticks. **c** Close-up view of the epitope region (cyan) interacting with Fab6D12 heavy chain (HC; yellow) and light chain (LC; orange). **d** The important epitope residue R105 inserts into an acidic pocket on Fab6D12, shown as surface representation and colored according to approximate net electrostatic potential (blue, positive; red, negative).

four basic epitope residues (Fig. 2b, c). The epitope is compact and relatively small, with ApoL1-NTD having a buried solvent-accessible surface of 690 Å².

*C. The Fab7D6:ApoL1-NTD and the Fab7D6:ApoL1-BH3 peptide complexes.* Crystals of the Fab7D6:ApoL1-NTD complex grew in space group C2 and diffracted to 1.91 Å (data collection and refinement statistics in Table 1). Surprisingly, the ApoL1-NTD formed a crystallographic domain-swapped dimer with one Fab7D6 bound to each protomer (Supplementary Fig 3a; electron density in Supplementary Fig. 2c). The ApoL1 protomer spans residues N91–K170 and is missing a 28-residue N-terminal segment, which comprises helices 1–2 of the four-helix core structure. Mass spectrometry analysis indicated that this was due to proteolytic cleavage between L88 and T89 during crystallization. As a result, the protomers in the domain-swapped dimer adopt a conformation that is quite different from the four-helix core structure. Additional attempts to crystallize the intact ApoL1-NTD with Fab7D6 did not produce crystals. SEC-MALS results clearly demonstrated that ApoL1-NTD and Fab7D6 formed a 1:1 complex of MW 65.0 kDa and not a dimer (Supplementary Fig. 1a). Therefore, we conclude that the dimer seen in the crystal structure represents an artifact. Upon further examination, we found that the central elements of helices 3–5 seen in the Fab3B6:ApoL1-NTD complex are preserved in the protomers of the

domain-swapped dimer (Supplementary Fig. 3b, c). Thus, the helices 1 and 2, which are missing in the protomers, seem to be critical for forming the four-helix core observed in the Fab6D12 and Fab3B6 co-crystal structures.

To ascertain that the Fab7D6-binding epitope region, which contains the BH3-like region, was not influenced by the artifactual dimer arrangement, we solved the structure of Fab7D6 bound to a peptide (ApoL1-peptide) encompassing the BH3-like region of ApoL1 (residues E152–H169) at a resolution of 2.16 Å (data collection and refinement statistics in Table 1). The entire peptide except the C-terminal His169 was resolved and a superposition with the corresponding region in the domain-swapped dimer showed that it adopts the same helix-turn-strand conformation (rmsd of 0.2 Å for 106 atoms) (Fig. 2d), which forms the epitope (E152–V168) with a buried solvent-accessible surface of 980 Å², the largest among the three Fab epitopes. The α-helix, which comprises most of the BH3-like region, starts at E152 and ends abruptly at residue D163 due to a turn initiated by G164 and it is followed by a 6-residue β-strand (electron density in Supplementary Fig. 2d). The Fab7D6-imposed turn-strand of the peptide is α-helical in the Fab3B6 structure (chain C), where the long α-helix comprises the entire BH3-like region (N154–Q166). However, the α-helix of the Fab7D6 structure superimposes very well (rmsd of 0.6 Å for 63 atoms) with that of the Fab3B6 structure (chain C) (Supplementary Fig. 3d).

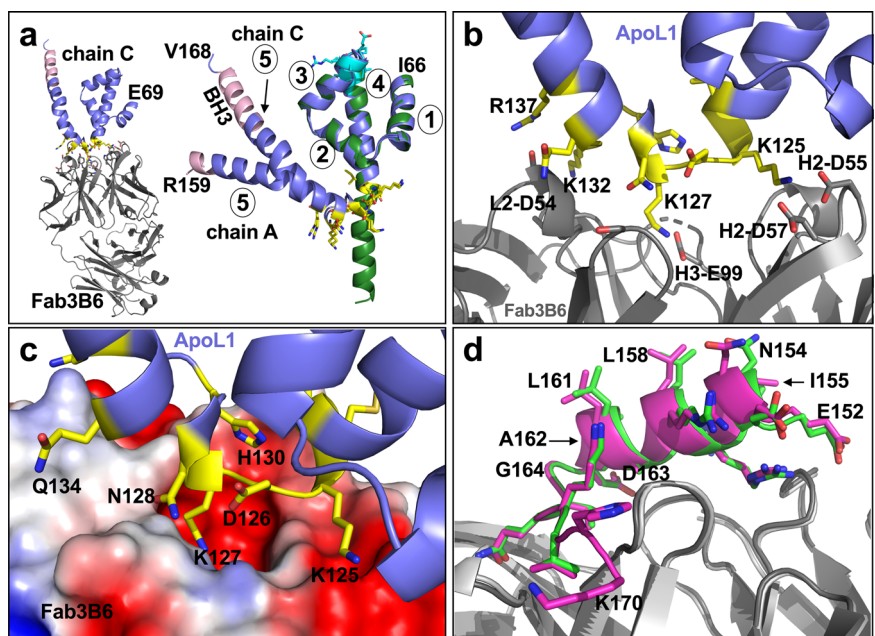

**Fig. 2 Structures of ApoL1-NTD in complex with Fab3B6 and of the BH3-like region bound to Fab7D6. a** Overview (left panel) of the complex of Fab3B6 (gray) with ApoL1-NTD (chain C in blue) with the N-terminal E69 indicated. The epitope region is in yellow. Right panel: the two different chains of the asymmetric unit, chain A (I66-R159) and chain C (E69-V168), are shown in blue with yellow epitope residues as sticks. Superposition with ApoL1-NTD from the Fab6D12 complex (dark green) shows that the four-helix cores (helices 1–4 are indicated) are virtually identical among the three structures, but they differ in that chains A and C of the Fab3B6 complex have a long, slightly bent helix 5 ending with the fully (chain C) or partially resolved (chain A) BH3-like region (in pink). **b** and **c** Close-up view of the epitope region (yellow) with the Fab as gray cartoon (**b**) or as surface (**c**), colored according to approximate net electrostatic potential (blue, positive; red, negative). The paratope surface is predominantly acidic, due to four acidic residues L2-D54, H2-D55, H2-D57, and H3-E99, which all engage in interactions with at least one of the four basic epitope residues K125, K127, K132, and R137. **d** The BH3-like region from the Fab7D6 co-crystal structures. Overlay of the BH3-like region from the Fab7D6:ApoL1-peptide complex (Fab in light gray, BH3 peptide in green) with that of the Fab7D6:ApoL1-NTD swapped dimer complex (Fab in dark gray; BH3-like region in magenta). The structures superimpose well, with the hydrophobic face of the amphipathic BH3-like helix pointing away from the Fab binding site.

**NMR solution structure of the ApoL1-NTD displays pH-dependent dynamics and reveals a helix-binding groove.** To determine whether the conformational states observed in the antibody-bound crystal structures are representative of the ApoL1-NTD structure in solution, we solved the NMR solution structure of ApoL1-NTD. Upon $^{15}$N-labeling ApoL1-NTD and measuring $^{15}$N-HSQC spectra, we identified a distinct pH dependence of the spectra. At pH 7.0, there were fewer peaks than expected and with substantial line-broadening, indicative of substantial conformational heterogeneity (Fig. 3a). Decreasing the pH to 6.0 and to 5.5 resulted in a substantial improvement in the spectra, with more peaks appearing and with more uniform peak intensities across the spectra (Fig. 3b, c). Further pH reduction to 5.0 led to minimal improvement in the spectrum (Fig. 3d) and, therefore, subsequent experiments were conducted at pH 5.5. Notably, the spectral pattern at pH 7.0 remains similar to that observed at lower pH conditions (Supplementary Fig. 4), suggesting that the lowest energy conformation sampled at pH 7.0 is largely similar to that sampled at pH 5.5, but that this state is more stable and/or less prone to aggregation at the lower pH. Even under the optimized pH condition at pH 5.5, regions of ApoL1-NTD remained dynamic, as indicated by considerable line-broadening still present in the spectrum (Fig. 3c). Utilizing $^{13}$C$^{15}$N-labeled ApoL1-NTD, we determined backbone and side chain assignments for the majority of the protein; however, residues Y136–L151 exhibited particularly severe line-broadening, precluding assignment of many of the atoms in this segment (Supplementary Fig. 4). We collected through-space NOE experiments to determine distance restraints and calculated a solution structure of ApoL1-NTD with a backbone rmsd among

ordered residues of 0.6 Å (Fig. 4a, see Table 2 for full statistics). In solution, α-helices 1–4 adopt a conformation nearly identical to that seen in the Fab6D12- and Fab3B6-bound crystal structures (Fig. 4b, c), diverging slightly at the C-terminus of α-helix 4, which adopts a straighter conformation in the antibody-bound structures. An alignment of the ApoL1-NTD α-helical segments of the different structures shows that the helix 1–4 boundaries are highly conserved between the NMR structure and the Fab3B6 antibody-bound structures (Fig. 5). However, in contrast to the Fab6D12 and Fab3B6 crystal structures in which the C-terminal helix projects away from the four-helix core (Fig. 4b, c), the NMR structure shows that the C-terminal helix 5 folds back and packs against α-helices 1 and 3 to form a five-helix bundle. This helix 5 is shorter than that in the Fab3B6 complex and it mainly comprises the BH3-like helix. The α-helices 4 and 5 are connected by a poorly defined dynamic linker corresponding to the most severely line-broadened residues in the NMR spectra (Fig. 4a, b).

In contrast to the Fab6D12 epitope, which adopts a nearly identical conformation as seen in the crystal structures, the Fab3B6 epitope is not formed and the Fab7D6 epitope is inaccessible, since the contact residues on helix 5 are buried (Fig. 4b). This implies that helix 5 is not firmly anchored to the four-helix core and is able to readily transition to a conformational state, represented by the Fab3B6 structure, that is compatible with Fab3B6 and Fab7D6 binding (Fig. 4c). During this transition, the helix 5 moves out of its binding groove and a portion of the disordered linker region forms the Fab3B6 epitope while another linker portion becomes part of the long helix 5 ending with a now accessible Fab7D6 epitope, i.e., the BH3-like helix (Fig. 4c). The cartoon in Fig. 4d illustrates these

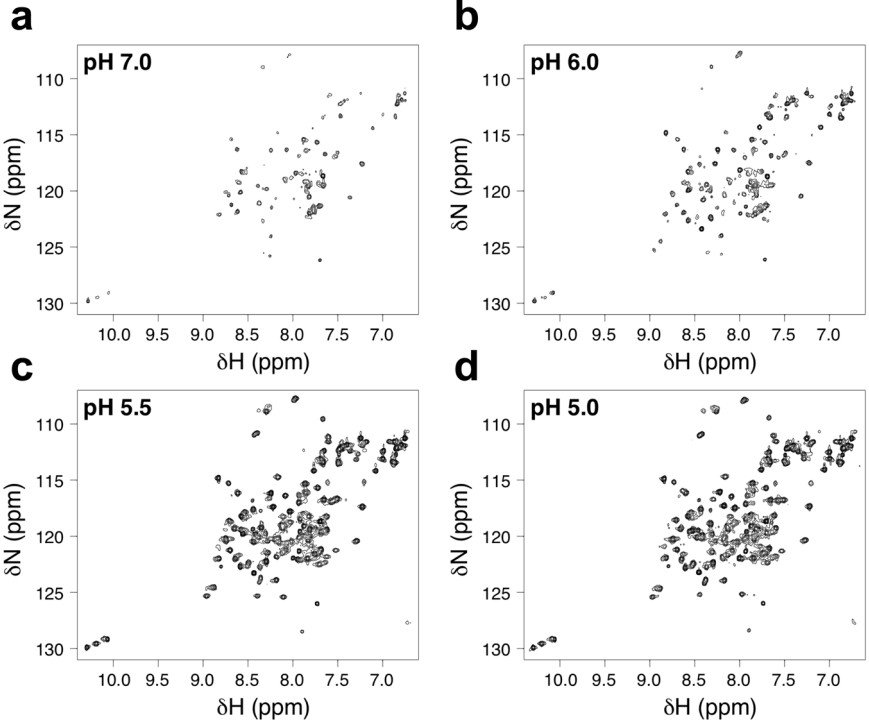

**Fig. 3 15N-HSQC spectra collected on ApoL1-NTD at different pH at 37 °C and all contoured to the same level. a** 15N-HSQC spectrum at pH 7.0. **b** 15N-HSQC spectrum at pH 6.0. **c** 15N-HSQC spectrum at pH 5.5. **d** 15N-HSQC spectrum at pH 5.0. At pH 7.0, the small number of peaks and nonuniform peak widths in the spectrum reflect a high degree of conformational heterogeneity. The increased peak numbers and more uniform peak intensities indicate that the NTD is stabilized into a single dominant conformational state upon reducing the pH to 5.5 or 5.0. A subset of residues continues to exhibit line broadening at pH 5.5 and 5.0, suggesting that parts of the NTD remain dynamic even in this stabilized state.

conformational states of helix 5, the "bound state" observed in the NMR structure and the "open state" seen in the Fab3B6 structure, where the helix 5 projects away from the four-helix core. In addition, helix 5 anchors the entire ApoL1-NTD to the membrane, as the C-terminal BH3-like region is connected through a short linker to two predicted transmembrane domains, the H-L-H segment[43].

The shallow groove harboring the BH3-like segment of helix 5 is composed of hydrophobic and a negatively charged regions (Fig. 4e). The hydrophobic surface is mainly formed by residues of helix 1 (F67, A71, F75) and helix 3 (F97, A101, plus linker L103). Some of these residues also engage in stabilizing the four-helix core conformation (F67, F75, F97), suggesting that a properly formed four-helix core is prerequisite for helix 5 docking. The main hydrophobic contacts are made by helix 5 residues L151, I155, and L158, which are arranged like a single Leu zipper heptad (LxxxIxxL binding motif), whereas R156 and R159 interact with the acidic portion of the groove (Fig. 4e). This raises the possibility that in the "open state" the vacated groove may accommodate the helix of a ligand (Fig. 4d), such as the Leu zipper containing C-terminal region of ApoL1 or ApoL3, which were recently shown to interact with the ApoL1-NTD[48].

**The BH3-like region of ApoL1 does not bind to pro-survival proteins**. A putative BH3 sequence of ApoL1 (Fig. 6a) was initially identified by Vanhollebeke and Pays[8] with an implied role in apoptosis. The herein identified structures of this BH3-like region superimpose well with the BH3-helix of the bona fide BH3-only protein Bid and the ApoL1 residues corresponding to the key Bcl-xL contact residues of Bid are in the same positions (Fig. 6b). However, a potentially problematic residue in ApoL1 is the bulky I155 at the first "s" position of the BH3-like motif $\Phi_1sxx\Phi_2xx\Phi_3sDz\Phi_4B$[55] (Fig. 6a and legend), which is always a

small residue (alanine, glycine, serine) in true BH3-only proteins[54]. Therefore, we synthesized 25-mer peptides spanning the ApoL1-BH3-like region, either as wildtype sequence (ApoL1-BH3), or as a mutated version with the small alanine at the "s" position instead of the isoleucine (ApoL1-BH3 (I155A)). These peptides, along with full-length ApoL1 (ApoL1) and ApoL1-NTD, were compared with Bid and Bim peptides (Fig. 6a) for binding to the pro-survival proteins Bcl-xL, Bcl-2, Bcl-w, Mcl-1, and Bfl-1 (also known as Bcl-2-related protein A1) by SPR. We also included the ApoL2-NTD (N2-T113) and a peptide encompassing the ApoL2-BH3-like motif (Fig. 6a) as a representative result of this study, Fig. 6c shows the binding kinetics for Bcl-xL. As expected, Bid and Bim bound with high affinity ($K_D$ 7.1 ± 0.9 nM and 3.3 ± 1.2 nM, respectively). However, the ApoL1-BH3 peptide did not interact with Bcl-xL and neither did ApoL1-NTD, nor ApoL1 (Fig. 6c). Similar results were obtained for Bcl-2, Bcl-w, Mcl-1, and Bfl-1. In each case, Bid and Bim showed strong binding to the pro-survival proteins as expected[63], whereas no binding interactions were detectable for ApoL1 proteins and peptides (Table 3). Similarly, the ApoL2-NTD and ApoL2-BH3 peptide did not bind to any pro-survival proteins (Table 3), which agrees with the inability of ApoL2 to induce apoptosis or autophagy[51]. The introduction of a bulky ApoL1 residue Ile at the first "s" position of Bim and Bid consistently decreased their binding affinity for the pro-survival proteins (Table 3), in agreement with the importance of a small residue at this position[54]. However, the reverse change in ApoL1 (mutant ApoL1-BH3 (I155A)) did not confer any detectable interaction (Fig. 6c and Table 3), suggesting that the lack of ApoL1 interaction with Bcl-2 proteins was not solely caused by the bulky Ile155 residue. These findings demonstrate that the ApoL1- and ApoL2-BH3-like regions are incapable of productively interacting with pro-survival Bcl-2 family proteins.

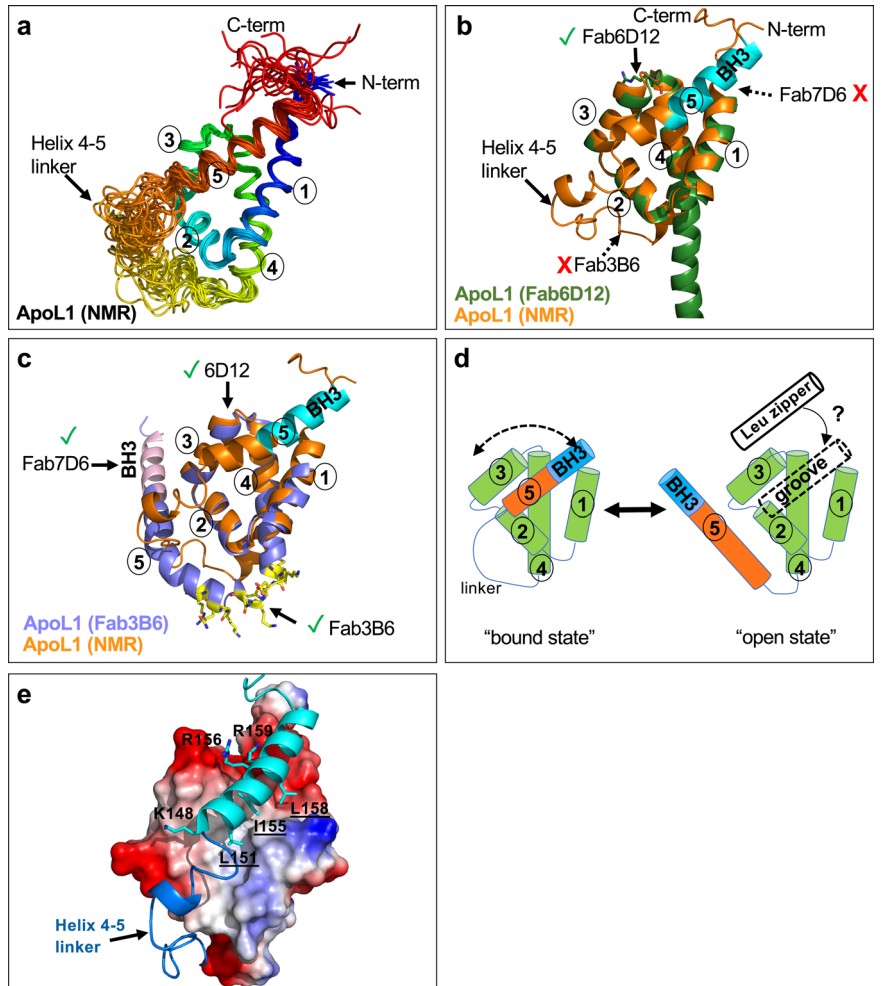

**Fig. 4 NMR solution structure of ApoL1-NTD at pH 5.5 reveals different conformational state of helix 5. a** Ribbon diagram of the 20 lowest energy ApoL1-NTD structures in rainbow colors (colored from blue to red (N- to C-term) according to colors of increasing wavelength). **b** Overlay of the lowest energy state from the NMR-calculated ensemble of ApoL1-NTD structures depicted in **a** (orange) with ApoL1-NTD from the Fab6D12 crystal structure (dark green). Both structures share the four-helix core conformation, but the NMR structure differs by having a disordered linker region (helix 4–5 linker) that ends with a well-structured helix 5 (BH3-like region in cyan), which packs against the four-helix core. In this structure, only the Fab6D12 epitope is formed, but not that of Fab7D6 or Fab3B6. **c** Overlay of the lowest energy state from the NMR-calculated ensemble of ApoL1-NTD structures depicted in **a** (orange) with ApoL1-NTD from the Fab3B6 crystal structure (chain C; blue with epitope residues in yellow). In contrast to the NMR structure (helix 5 in the "bound state"), the crystal structure shows that helix 5 (ending with the BH3-like helix; pink color) projects away from the four-helix core; in this structure, all three antibody epitopes are formed and are accessible. **d** Model of the two conformational states of helix 5. The "bound state" (NMR structure) in which helix 5 (orange with C-terminal BH3-like region in blue) is placed in a groove formed by helices 1 and 2 is in equilibrium with the "open state" (Fab3B6 structure) in which the helix 5 projects away from the four-helix core and is elongated due to the addition of linker residues into the helix. The vacated groove in the "open state" could be occupied by other interactors, such as the C-terminal Leu zipper helix ("Leu zipper" as cylinder) of ApoL family members. **e**. Surface representation of the four-helix core (surface is colored according to approximate net electrostatic potential: blue, positive; red, negative) with helix 5 (cyan) and the helix 4–5 linker (blue) as cartoon. The shallow groove that harbors helix 5 (mainly its BH3-like portion) is composed of a hydrophobic and a negatively charged portion. The main hydrophobic contacts are made by helix 5 residues L151, I155, and L158 forming the LxxxIxxL motif (cyan sticks; underlined residues), whereas K148, R156, and R159 engage with the acidic portion (cyan sticks).

**The four-helix core conformation is a structural motif shared among ApoL family members.** In a previous study, we found that Ab6D12 recognized other ApoL family members, including ApoL2[64]. An alignment of the ApoL1-NTD with the corresponding NTD of ApoL2 (N2-T113) showed that the Ab6D12 epitope, including the important R105 residue (R46 in ApoL2) is highly conserved (Fig. 5). Therefore, we used Fab6D12 to determine the structure of the ApoL2-NTD, which like the ApoL1-NTD ends a few amino acids upstream of a predicted membrane-spanning region (DAS transmembrane prediction server: https://tmdas.bioinfo.se/DAS/index.html;[65]) and did not require detergents for purification and crystallization. Crystals grew in space

group C222₁ and diffracted to 2.15 Å (data collection and refinement statistics in Table 1). The complex superimposed well with that of the Fab6D12:ApoL1-NTD complex (Fig. 7a), displaying a virtually identical four-helix core structure (Fig. 7b) (rmsd of 0.5 Å for 319 atoms) with almost identical helical boundaries as found in the ApoL1 structures (Fig. 5). Similarly, the four N-terminal residues (N2–S5) are disordered. However, helix 4 is not further extended as seen in the Fab6D12:ApoL1-NTD complex, but this entire segment (H62–T113) is disordered, due to differences in crystal packing. The epitope conformation is fully preserved and Fab6D12 engages in similar interactions with ApoL2-NTD as seen with ApoL1-NTD, including the key

interaction of the R46 residue (R105 in ApoL1) with the acidic pocket of Fab6D12 (Fig. 7c, electron density in Supplementary Fig. 2e). The only amino acid change in the epitope is residue D47 (Figs. 5, 7b), which nonetheless forms an H-bond with the Fab CDR-L3 N92 residue as does the corresponding N106 residue in ApoL1. These results demonstrate that ApoL1 and ApoL2 share the exact same four-helix conformation of their NTDs.

A structure search with DALI (http://ekhidna2.biocenter. helsinki.fi/dali/;[66]), using the four-helix core sequence E69–K125 of the ApoL1-NTD (chain C) from the Fab3B6 structure as a search model, did not identify any structurally similar proteins. Therefore, the four-helix core

conformation we determined for APOL1-NTD thus far is only found in ApoL family members. The DALI search included the structure of the pore-forming domain of colicin A (colicin A-PFD)[67], an antibacterial toxin that forms voltage-dependent ion channels, which was proposed to have a similar fold as the ApoL1-NTD[39]. Like ApoL1-NTD, the colicin A-PFD is mainly composed of α-helices, but they are arranged quite differently (Supplementary Fig. 5), suggesting that the ApoL1-NTD may not be a PFD. A comparison of colicin A (PDB 1COL, chain A) with the four-helix core of ApoL1-NTD (Fab3B6 structure) using PDBefold (Protein structure comparison service PDBeFold at European Bioinformatics Institute (http://www.ebi.ac.uk/msd-srv/ssm), authored by E. Krissinel and K. Henrick[68]) showed low scores ($Q = 0.1$, $P = 0.0$, and $Z = 3.3$, 2% sequence identity), further corroborating the dissimilarity of these two proteins. Moreover, the two proteins also differ in their pH dependency: acidic pH conditions favor the structural stabilization of the ApoL1-NTD according to our solution NMR results, whereas the colicin A-PFD undergoes a pronounced structural destabilization at low pH values[69,70]. We note that in comparison to the colicin A-based model of ApoL1[39], our ApoL1-NTD construct is missing the C-terminal sequence T173–W235. However, this C-terminal portion mainly encompasses the predicted H-L-H transmembrane segment[43], which is not anticipated to contribute to the surface-exposed NTD structure[61]. Furthermore, Bcl-2 family members, which were proposed to share structural similarity with ApoL1[8,17], do not display the four-helix fold of ApoL1 and ApoL2, as exemplified by a structural comparison with Bcl-x$_L$ (Supplementary Fig. 5).

The ApoL1 and ApoL2 structures show that the Fab6D12 epitope is only formed in the context of a properly folded four-helix core, since the epitope (helix 3/linker/helix 4) requires stabilization by helices 1 and 2. Therefore, we used Ab6D12 as a probe to find out whether the four-helix core motif is also present in other ApoL family members. Immunoprecipitation experiments of lysates from transfected COS cells expressing ApoL1-6 showed that Ab6D12 bound to ApoL1–4, albeit only with a weak interaction to ApoL4 (Supplementary Fig. 6a). Similarly, immunofluorescence staining demonstrated that Ab6D12 recognized ApoL1–4, but not ApoL5 and 6 (Supplementary Fig. 6b), suggesting that ApoL3 and ApoL4 share the same four-helix conformation with ApoL1 and ApoL2. In support, the NTDs of ApoL3 and ApoL4 have predicted α-helices (JPred4)[71] with boundaries that closely correspond to the helices 1–4 observed in

**Table 2 Structural statistics for the NMR assignments and solution structure of the ApoL1-NTD.**

|  | ApoL1-NTD |
|---|---|
| **NMR distance and dihedral constraints** | |
| Distance constraints | |
| Total NOE | 2022 |
| Intra residue | 482 |
| Inter residue | 1540 |
| Sequential ($\|i - j\| = 1$) | 558 |
| Medium range ($\|i - j\| < 4$) | 629 |
| Long range ($\|i - j\| > 5$) | 353 |
| Hydrogen bonds | 52 |
| Total dihedral angle restraints | 172 |
| φ | 86 |
| ψ | 86 |
| **Structure statistics** | |
| Distance violations per structure | |
| 0.1–0.2 Å | 17.4 |
| 0.2–0.5 Å | 2.5 |
| >0.5 Å | 0 |
| r.m.s dihedral angle violation (°) | 0.6 |
| r.m.s. distance violation (Å) | 0.02 |
| Max. dihedral angle violation (°) | 6.7 |
| Max. distance constraint violation (Å) | 0.38 |
| Average pairwise r.m.s. deviation[a] (Å) | |
| Heavy | 0.9 (2.9) |
| Backbone | 0.6 (2.3) |

[a]Pairwise r.m.s. deviation was calculated among 20 refined structures. Ordered residues 61–126 and 150–163 are reported, with all-residue r.m.s. deviations reported in parentheses.

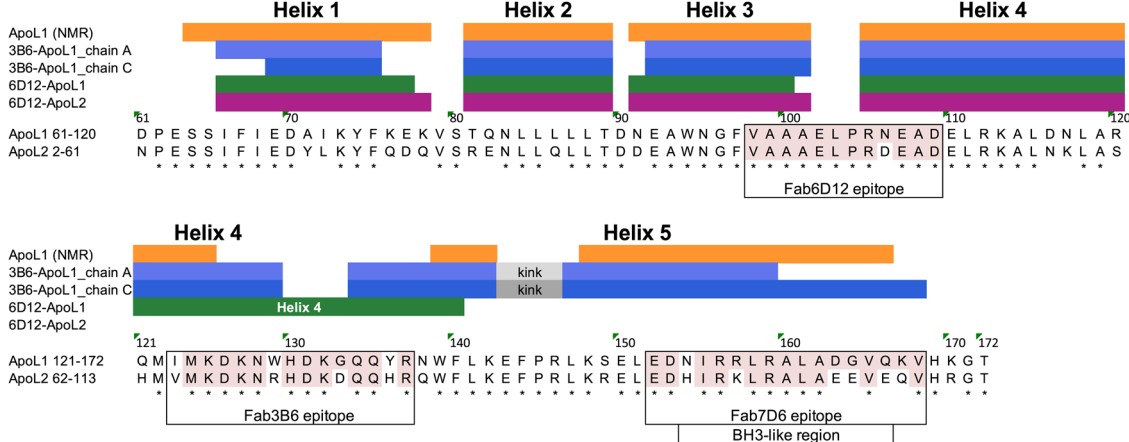

**Fig. 5 Comparison of helix boundaries among the determined ApoL1 and ApoL2 structures.** ApoL1-NTD and ApoL2-NTD sequences are aligned with the Fab epitopes in boxes and the conserved epitope residues highlighted in pink color. The helix boundaries, determined by using CCP4 and ProCheck, are shown for the four crystal structures and the ApoL1 NMR structure. The kink in the long ApoL1-helix 5 in the two Fab3B6 structures, due to the helix-breaking P145 residue, is shown as gray box.

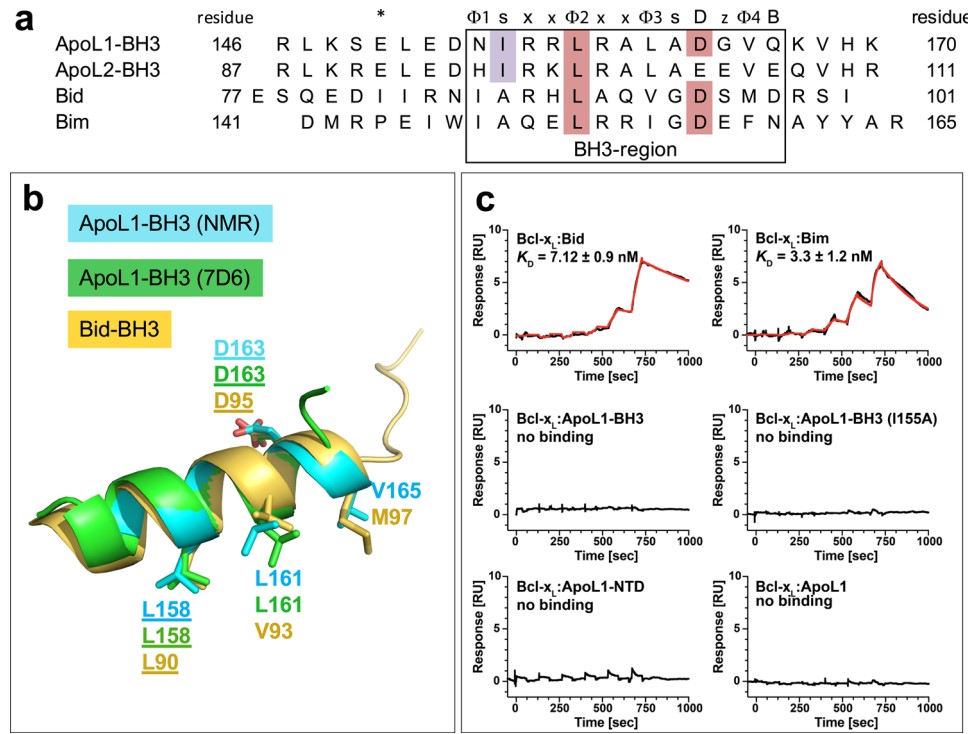

**Fig. 6 The BH3-like region of ApoL1 does not bind to Bcl-2 family members. a** Alignment of the 25-mer peptides encompassing the BH3-like region of ApoL1 and ApoL2 and the BH3-only motif of Bid and Bim. The BH3-only motif $\Phi_1sxx\Phi_2xx\Phi_3sDzs\Phi_4B$ contains four hydrophobic residues $\Phi_1-\Phi_4$, small residues "s" (e.g. Gly, Asp, Ser), an acidic residue "D," and an H-bond acceptor "B." Highlighted are the potentially problematic Ile residue of ApoL1 and ApoL2 (purple) and the hallmark residues Leu ($\Phi_2$) and Asp (D) (both in red), which engage in binding to Bcl-2 proteins. The asterisk indicates the position of E150 of the ApoL1-E150/I228/K255 haplotype. **b** The BH3-like helix of ApoL1 from the NMR structure (cyan) and the Fab7D6:ApoL1-BH3 peptide structure (green) superimpose well with the Bid BH3-only helix (gold; from PDB 4QVE). Key residues including leucine ($\Phi_2$, underlined) and aspartic acid (D, underlined) are shown as sticks. **c** Representative results from SPR experiments showing single-cycle kinetics of the binding interactions between immobilized Bcl-$x_L$ and its ligands Bid ($K_D$ 7.1 ± 0.9 nM) and Bim ($K_D$ 3.3 ± 1.2 nM) and the lack of binding to the ApoL1-BH3 peptide, the ApoL1-NTD and ApoL1 (up to 3 μM concentration). The change of the ApoL1 residue I155 to the canonical small residue alanine (ApoL1-BH3 (I155A)) did not confer any detectable interaction (up to 3μM concentration). All experiments were done in triplicate and repeated at least three times.

**Table 3 ApoL1-BH3 peptides and purified proteins show no interaction with Bcl-2 family proteins.**

| Analyte/ligand | Bcl-w $K_D$ ± S.D. (nM) | Bcl-$x_L$ $K_D$ ± S.D. (nM) | Mcl-1 $K_D$ ± S.D. (nM) | Bfl-1 $K_D$ ± S.D. (nM) | Bcl-2 $K_D$ ± S.D. (nM) |
|---|---|---|---|---|---|
| Bid | 24.2 ± 4.8 | 7.12 ± 0.9 | 12.8 ± 3.1 | 0.31 ± 0.1 | 96.7 ± 8.2 |
| Bid (A87I) | 65.2 ± 12.5 | 28.5 ± 8.0 | n.b. | 1430 ± 200 | 158 ± 10.2 |
| Bim | 5.5 ± 0.2 | 3.3 ± 1.2 | 0.37 ± 0.2 | 0.43 ± 0.1 | 14.2 ± 0.1 |
| Bim (A149I) | 23.1 ± 1.7 | 12.2 ± 1.3 | 0.82 ± 0.1 | 18.8 ± 0.9 | 115 ± 11.3 |
| ApoL1-BH3 | n.b. | n.b. | n.b. | n.b. | n.b. |
| ApoL1-BH3 (I155A) | n.b. | n.b. | n.b. | n.b. | n.b. |
| ApoL1-NTD | n.b. | n.b. | n.b. | n.b. | n.b. |
| ApoL1 | n.b. | n.b. | n.b. | n.b. | n.b. |
| ApoL2-BH3 | n.b. | n.b. | n.b. | n.b. | n.b. |
| ApoL2-NTD | n.b. | n.b. | n.b. | n.b. | n.b. |

Kinetic constants were determined by surface plasmon resonance and represent the average ± S.D. of at least three independent experiments. No binding (n.b.) was detected up to 3μM final concentration.

the crystal structures of ApoL1 and ApoL2 (Supplementary Fig. 7). An alignment of the NTD sequences of ApoL1–5 shows that many of the important hydrophobic residues that are buried and stabilize the four-helix arrangement are fully conserved among ApoL1–4, but less so in ApoL5 (Supplementary Fig. 7). The high similarity between ApoL1 and ApoL family members 3 and 4 allowed us to generate structural models of the four-helix region of ApoL3 (residues F73–T132) and ApoL4 (residues F29–P88) using the Rosetta homology modeling server[72] with the ApoL1 NMR structure as a template. The predicted models can be superimposed with the four-helix core of the ApoL1 NMR

structure, having identical helix boundaries and linker regions (Supplementary Fig. 8a, b). The side chains of all the conserved hydrophobic residues (Supplementary Fig. 7), which project toward the interior and stabilize the four-helix conformation are virtually superimposable (Supplementary Fig. 8c). Moreover, consistent with the immunoprecipitation and immunofluorescence results (Supplementary Fig. 6), the Fab6D12 epitope is preserved in the ApoL3 and ApoL4 models, except for the ApoL4-E74 residue (Supplementary Fig. 8d). The most common rotamer of ApoL4-E74 when superimposed on ApoL1-N106 shows that it adopts a conformation, which would clash with the

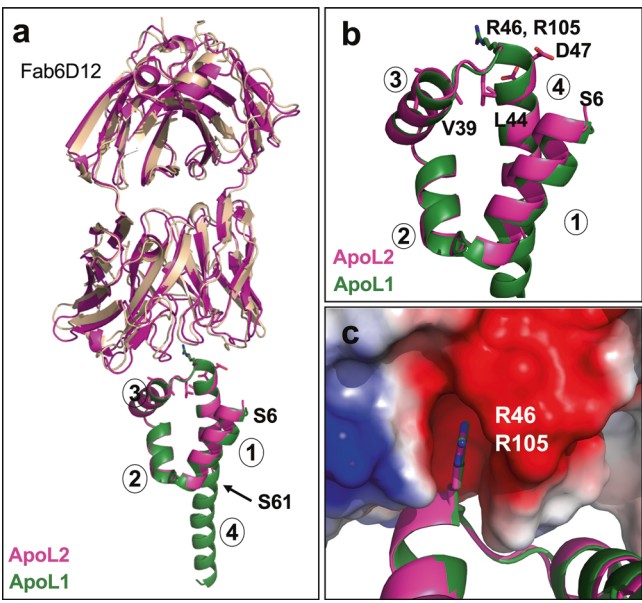

**Fig. 7 Structure of the Fab6D12:ApoL2-NTD complex. a** The ApoL2-NTD (residues S6–S61; magenta) in complex with Fab6D12 (magenta) superimposed with ApoL1-NTD (dark green) in complex with Fab6D12 (brown) with key epitope residues as sticks. **b** Close-up view showing the almost identical structure of the four-helix core regions (helices 1–4 are indicated) between ApoL2 (magenta; starting with residue S6) and ApoL1 (dark green). ApoL2 epitope residues are shown as sticks; the key contact residue R46 superimposes with the corresponding R105 residue of ApoL1 (dark green stick). **c** The key contact residue of ApoL2 (R46, magenta stick) and of ApoL1 (R105, green stick) superimpose perfectly and insert into the identical acidic pocket of the Fab6D12, which is shown in surface representation and colored according to approximate net electrostatic potential (blue, positive; red, negative).

main chain of S30 from the light chain of Fab6D12 (Supplementary Fig. 8e, f), which could explain the observed weak interaction of AbD12 with ApoL4 (Supplementary Fig. 6).

The lack of Ab6D12 immunoreactivity with ApoL5 and ApoL6, which are the most distant relatives of ApoL1[3], suggests that their NTDs do not adopt the four-helix conformation or that the Ab6D12 epitope is no longer preserved. The latter may be the case for ApoL5 (Supplementary Fig. 9). Therefore, we are unable to make a firm conclusion regarding the presence or absence of the four-helix conformation in ApoL5. ApoL6 has a deletion at its N-terminus encompassing helices 1 and 2 and thus will be incapable of forming the four-helix core. In contrast to Ab6D12, the binding of Ab7D6 does not require the four-helix fold, since a peptide encompassing the epitope region is recognized by Fab7D6 (Fig. 2d). However, a sequence alignment shows that many important contact residues are not well preserved among other family members, including charged residues and the helix-capping glycine residue (Gly164 in ApoL1; Supplementary Fig. 9), agreeing with the antibody's specific recognition of ApoL1 in cell-based systems (Supplementary Fig. 6). The Ab3B6 epitope is formed by a complex helix-turn-helix segment and may partially depend on a properly formed four-helix core. In agreement, this antibody recognized ApoL members 1–3 and by immunofluorescence also weakly ApoL4 (Supplementary Fig. 6), all four of which form the four-helix core motif.

## Discussion

The described structures of the ApoL1-NTD comprise the soluble portion of the extended transmembrane-containing N-terminal region, which was shown to be critical for ApoL1 innate immune function[39] and for podocyte cytotoxicity elicited by the ApoL1 risk variants[25]. The ApoL1-NTD adopts a four-helix fold, which differs from the classical four-helix bundle[62,73]. This ApoL1-NTD conformation is different from previous models predicting a structure that is related to colicin-A or Bcl-2 family members[39]. Therefore, there is no apparent structural basis to suppose that the pore-forming mechanism of ApoL1 has any relationship to that of colicin A. Recent findings by Schaub et al.[43] demonstrate that the ApoL1 pore-containing residues are actually not located within the N-terminal portion of ApoL1, but instead in a C-terminal transmembrane segment preceding the Leu zipper region. This leads to the conclusion that the NTD is not part of the pore. However, this does not preclude a potential role of the NTD in the formation of ApoL1 ion channels. The NTD is immediately followed by a predicted transmembrane segment, the H-L-H region, which was suggested to serve as an anchor for membrane insertion at low pH condition[43], an essential first step in ApoL1 channel formation[36,40,43,44]. Therefore, we consider the possibility that the NTD may cooperate with the adjoining H-L-H region in this process, since the low pH conditions at which membrane insertion occurs also favor the structural stabilization of the isolated NTD as determined by solution NMR. Whether and exactly how the NTD might facilitate H-L-H insertion remains unproven and unknown at this point. In addition to the cellular form, ApoL1 is also associated with the lipid monolayer of circulating HDL particles[74,75], which serve as carriers to deliver ApoL1 to invading trypanosomes in the innate immune response. The question arises as to whether the NTD may adopt a similar structure in this different HDL lipid environment. A comprehensive epitope mapping study with a large panel of anti-ApoL1 antibodies demonstrated that all thirteen NTD-binding antibodies, including the three antibodies used in the present study, bound to podocyte and CHO cell surface ApoL1[61], consistent with a fully exposed and accessible NTD. However, only five antibodies recognized HDL-associated ApoL1, suggesting that many NTD regions are inaccessible and/or that conformation-dependent epitopes are no longer formed. One explanation could be that the hydrophobic residues of the five amphipathic NTD helices engage in interactions with the lipid monolayer of HDL particles. These hydrophobic residues are buried in the four-helix core structure and are important to stabilize this fold and, therefore, the HDL-bound NTD would be expected to adopt a conformational state(s) that is quite different from that seen in our structures. This interpretation remains speculative and more accurate insights can only be gained by structural studies with lipid-associated NTD or full-length ApoL1.

The NTD construct used herein ends with the BH3-like region of helix 5 just a few amino acids upstream of the transmembrane H-L-H segment. Unlike the stable four-helix core, which is virtually identical among the different structures reported herein, this C-terminal helix 5 is able to transition among different conformational states, the "bound" and the "open" states. The "bound" state may represent a minor form, since it is not compatible with the binding of antibodies 7D6 and 3B6, both of which bind to ApoL1 with high affinity and readily recognize cell surface ApoL1 on podocytes and CHO cells[61]. Thus, the available "open" state may allow the groove, which is formed by the four-helix core, to engage in interactions with helical ligands that share the LxxxIxxL binding motif of helix 5, such as the C-terminal Leu zipper regions of ApoL family members. In support of this, Uzureau et al.[48] used an ApoL1 N-terminal construct that roughly corresponds to the NTD used herein to demonstrate that it interacts in-cis with the ApoL1 C-terminal region and in-trans with ApoL3 to regulate neuronal calcium sensor-1 and effecting actomyosin organization in podocytes.

The BH3-like region, which is a part of the long ApoL1-helix 5, is implicated in promoting cell death, but the functional significance of the BH3-like motif has remained contentious. Four of our structures show a fully resolved BH3-like region adopting an α-helix with the hallmark residues Leu158 and Asp163 oriented as found in true BH3-only proteins. However, a comprehensive SPR study failed to demonstrate any binding interaction with the five Bcl-2 homologs Bcl-x$_L$, Bcl-2, Bcl-w, Mcl-1, and Bfl-1, even after mutating the putatively problematic Ile155, which is conserved in the BH3-like regions of the human and mouse ApoL family[7,8], to the canonical alanine residue. Therefore, we conclude that ApoL1 belongs to the group of noncanonical BH3-containing proteins[53], and that it does not mediate cytotoxicity by direct interaction with pro-survival Bcl-2 family proteins, in agreement with conclusions drawn by others[56,57]. While we cannot rule out the possibility that the ApoL1-BH3-like region directly interacts and activates the pro-apoptotic effector proteins Bax and Bak, akin to the BH3-only proteins Bid and Bim[76], the fact that >95% of ApoL1 transcripts in podocytes encode secretory isoforms[64] means that the NTD of most ApoL1 proteins are physically separated by a bilayer from cytoplasmic Bax and Bak. A noncanonical function of the ApoL1-BH3-like region is further suggested by the finding that baboon ApoL1 is able to lyse trypanosomes, but lacks the hallmark residue Asp in its BH3-like region[19]. Moreover, the ApoL1-BH3-like region was shown to promote membrane permeabilization and lysis of trypanosomes, even if the hallmark Asp residue was mutated[77] and even though the *T. brucei* genome lacks Bcl-2 family members[78,79]. Therefore, the function of the BH3-like region requires neither a true BH3 motif, nor interaction with Bcl-2 family proteins. Because ApoL1 lacks a functional BH3 motif, the term "BH3-like" rather than "BH3 domain" or "BH3 motif" seems more fitting.

Interestingly, the E150 residue of the E150/I228/K255 haplotype found in Africa[42], is situated just a few amino acids upstream of the BH3-like region. ApoL1-G1 and ApoL1-G2-mediated toxicity in HEK-293 cells[80] and podocytes (N. Gupta and S. J. Scales, unpublished) is greater for the E150/I228/K255 haplotype than the more common K150/I228/K255 haplotype[42], suggesting that the E150K SNP may somehow increase ApoL1 activity in this context. Our structural studies were carried out with the E150 variant of ApoL1-NTD and in the Fab3B6 crystal structure, as well as in the solution NMR structure, we find this surface-exposed residue situated in the extended helical region of the BH3-like region. The K150 variant, as found in the K150/I228/K255 haplotype, is predicted to neither alter the overall structure of the "open form" of ApoL1-NTD (Fab3B6 complex), nor the "closed form" (NMR structure).

In addition to ApoL1 and ApoL2, the four-helix fold may be shared by the ApoL3- and ApoL4-NTD, based on sequence homology and reactivity with the conformation-specific Ab6D12. Supporting these data, structural models of the ApoL3- and ApoL4-NTD show that the highly conserved hydrophobic residues, which stabilize the four-helix conformation, are in the same positions as found in ApoL1 and ApoL2. Another common feature shared by ApoL1–4 is that the conserved helix 5 is followed by predicted transmembrane domains (DAS transmembrane prediction server: https://tmdas.bioinfo.se/DAS/index.html;[65]), suggesting that their NTDs are similarly anchored to a lipid bilayer and indeed they appear membrane-bound by immunofluorescence experiments (Supplementary Fig. 6b).

ApoL proteins are upregulated by various pro-inflammatory cytokines[6,7] and, therefore, immune-related functions have been suggested[6–8], although little is known except for the well-described innate immune function of ApoL1. The herein reported first structural insights into members of the ApoL family identified an ApoL family-specific domain motif, which may imply a shared function in immune defense. These findings should expedite the quest toward a more comprehensive structural and functional understanding of this intriguing protein family.

## Methods

### Expression and purification of anti-ApoL1 Fabs 6D12, 7D6, and 3B6.
The variable sequences of the mouse Fab light and heavy chains of Ab6D12, 3B6, and 7D6 (previously reported as 3.6D12, 3.3B6, and 3.7D6, respectively[61,64]) were amplified by PCR using overlapping oligonucleotides designed for restriction-independent cloning and then humanized using the anti-HER-2 Fab (PDB 1FVD) with the human C$_L$ kappa subgroup I and C$_H$ subgroup III domains[81]. The product was subcloned into *E. coli* expression plasmid AEP1 and transformed into expression strain 64B4. The resulting Fab proteins were secreted into the periplasm. The *E. coli* cell pellet was lysed using a cell disrupter (Microfluidics Corp.) and the lysate was clarified by centrifugation. The Fabs were purified from the supernatant by standard protein G column affinity techniques, cation exchange chromatography using SP Sepharose, and finally size-exclusion chromatography using a Superdex 75 16/60 column. The final protein buffer was 0.15M NaCl, 20mM Tris-HCl, pH 7.5.

### Expression and purification of ApoL1, ApoL1-NTD, and ApoL2-NTD.
The full-length ApoL1 (ApoL1, RefSeq "G4," NM_003661, with E150/M228/R255) containing an N-terminal flag-tag was expressed and purified as described[61] and was used for SPR and biolayer interferometry binding studies. The purified ApoL1 was functionally active in trypanosome lysis assays carried out as described previously[61]. Expression of ApoL1-NTD with an N-terminal flag-TEV cleavage sequence tag, ApoL2-NTD (ApoL2, Genbank NP_112092.2) with an N-terminal His6-TEV cleavage sequence tag and of ApoL2-NTD with an N-terminal flag-TEV tag was performed in Sf9 insect cells in a 4l Wave® bioreactor with baculovirus medium for 72 h. The frozen cell pellets were thawed into 0.25M NaCl, 50mM HEPES pH 7.5. The mixture was homogenized and lysed using a Microfluidizer (Microfluidics Corp.). The suspension was clarified by ultracentrifugation and the supernatants processed as follows: (i) the supernatant of flag-tagged ApoL2-NTD was purified using anti-flag resin according to the procedure described for full-length ApoL1[61] but without the use of detergent. (ii) The supernatant of flag-tagged ApoL1-NTD was loaded onto an anti-flag resin column (G-flag, 5 ml) equilibrated with 0.25M NaCl, 50mM HEPES pH 7.5 followed by elution with 1 mg/ml of flag peptide in 0.25M NaCl, 50mM HEPES pH 7.5. For X-ray crystallographic studies, the flag-tag of ApoL1-NTD was removed by digestion with TEV protease[82] overnight at 4 °C. The proteins were loaded onto a S75 (16/60) size-exclusion column in 0.15M NaCl, 25mM HEPES pH 7.5 and the ApoL1-NTD peak fractions were pooled for structural and biochemical studies. (iii) The supernatant of His-tagged ApoL2-NTD was loaded onto a Ni resin column (GE, 5 ml) equilibrated with 0.25M NaCl, 50mM HEPES pH 7.5, 50mM imidazole, and eluted with 300mM imidazole in 0.25M NaCl, 50mM HEPES pH 7.5. The protein peak corresponding to ApoL2-NTD was applied onto a S75 (16/60) size-exclusion column and eluted in 0.25M NaCl, 25mM HEPES, pH 7.5. For X-ray crystallographic studies, the His-tag of ApoL2-NTD was removed by digestion with TEV protease overnight at 4 °C and the processed ApoL2-NTD was further purified on a S75 size-exclusion column. All samples were then analyzed by SDS-PAGE and mass spectrometry. After TEV protease cleavage, both the ApoL1-NTD and ApoL2-NTD N-termini have two additional residues, Gly-Ser, which are part of the TEV protease cleavage sequence ENLYFQ/GS and were retained after treatment with TEV protease.

For NMR studies, uniform $^{15}$N-labeled or $^{13}$C$^{15}$N-labeled ApoL1-NTD protein was generated by induction of BL21(DE3) *E. coli* in either $^{15}$N or $^{13}$C$^{15}$N minimal media with 0.5mM IPTG for 4 h at 37 °C. Cells were lysed using a microfluidizer (Microfluidics Corp.), cleared, and loaded onto a prepacked Ni fast flow column (GE, 5 ml) equilibrated with 0.15M NaCl, 50mM Tris pH 7.5, 10% glycerol and then eluted with 300mM imidazole, 0.15M NaCl, 50mM Tris pH 7.5, 10% glycerol. The protein was then loaded directly onto a S75 (16/60) size-exclusion column and eluted in 0.25M NaCl, 25mM Tris, pH 7.5. The protein peak corresponding to ApoL1-NTD was pooled and digested with TEV protease overnight at 4 °C, analyzed by SDS-PAGE and mass spectrometry, and loaded onto a S75 (16/60) size-exclusion column and eluted in 25mM Tris pH 7.0, 150mM NaCl ($^{15}$N-labeled) or 25mM NaCl, 25mM MES pH 5.5, 1mM EDTA ($^{13}$C$^{15}$N-labeled).

### Crystallization and X-ray structure determination.
Complexes of anti-ApoL1 Fabs with ApoL1-NTD and of Fab6D12 with ApoL2-NTD were produced by adding 1.5 molar excess of NTD to purified Fabs. The complexes were purified using a Superdex-200 size-exclusion column in 0.15M NaCl, 25mM Tris pH 8.0. The complexes were concentrated to about 10 mg/ml.

The optimized crystallization condition for the Fab6D12:ApoL1-NTD complex was 20% PEG 3350, 0.1M ammonium sulfate, 0.1M MES, 14mM sodium cholate, pH 6.0. Crystals were preserved using 30% (w/v) PEG 3350, 0.15M ammonium sulfate, 0.1M MES, 14mM sodium cholate, pH 6.0 as a cryo-buffer and sudden immersion into liquid nitrogen. Data were collected at APS 22ID and 100 K/−173.5 °C at a wavelength of 0.98 Å and processed using XDS[83] to 2.26 Å in the

space group P4₃. Molecular replacement was carried out using Phaser[84] and humanized anti-HER2 Fab (PDB accession code 1FVD) as the search probe. Two Fab molecules were found in the asymmetric unit and the space group determined to be P4₃. After rigid-body refinement, clear helical density could be seen in the mFo-DFc electron density maps. Phenix.refine[85] simulated annealing was implemented. Rounds of manual fitting used COOT[86]. Phenix.refine with NCS refinement allowed the ApoL1-NTD to be registered using the Fab contacts as a point of reference. Higher resolution data to 2.03 Å were processed using Staraniso[87] from the Buster software package[88] and used for the final refinement Phenix.refine (Table 1). The final model contained 98.72% of the residues within favored regions of the Ramachandran plot and 0% of outliers. Final Clashscore: 0.62.

Fab6D12:ApoL2-NTD complex crystals from 20% PEG 3350, 0.2M di-hydrogen phosphate condition were preserved with 25% glycerol and suddenly immersed in liquid nitrogen. Data were collected at ALS 5.0.2 and 100 K/−173.5 °C at a wavelength of 0.98 Å and processed with XDS[83] to 2.89Å resolution. Molecular replacement was performed using the Fab6D12 from the Fab6D12-ApoL1-NTD complex in the space group C222₁. One complex was found in the asymmetric unit. The same data were re-reduced using DIALS[89] and scaling with Staraniso[87] to produce a 2.15Å dataset. The anisotropically scaled data were used for the final refinement with Buster[88] (Table 1). The final model contained 97.0% of the residues within favored regions of the Ramachandran plot and 0.21% of outliers. Final Clashscore: 0.13.

Fab3B6:ApoL1-NTD complex crystals from 40% MPD, 5% PEG 8000, 0.1M sodium cacodylate pH 6.5 were suddenly immersed in liquid nitrogen. Data were collected at ALS 5.0.2 and 100 K/−173.5 °C at a wavelength of 0.98 Å and processed to 2.4Å resolution in the space group P1. The light variable chain from mouse PDB 2VL5 was used along with separate variable and constant regions of a humanized anti-HER2 Fab (PDB accession code 1FVD) as search probes. Molecular replacement was performed using Phaser[84]. Refinement[85] and manual fitting of side chains[86] rectified the Fab3B6 sequence. From the 2mFo-DFc maps the helices appeared to have the same core fold as the Fab6D12-NTD structure, but with the C-terminal helix orientated 180° away starting at residue K125. This difference from the prior structure is central to presenting the Fab3B6 epitope. The final model contains two complexes per asymmetric unit. Refinement employed BUSTER[88] with NCS and TLS restraints. Final refinement used anisotropically scaled data from Staraniso[87] to 1.86 Å and Phenix.refine[85] (Table 1). The final model contained 96.76% of the residues within favored regions of the Ramachandran plot and 0.19% of outliers. Final Clashscore: 1.37.

Fab7D6:ApoL1-NTD complex crystallized using 15% (w/v) PEG 4000, 0.15M ammonium sulfate, 0.1M MES, pH 6.0. Crystals were preserved using 30% w/v PEG 3350 and were quickly immersed into liquid nitrogen. Data were collected at ALS 5.0.2 and 100 K/−173.5 °C at a wavelength of 1 Å and processed to 1.91 Å. The structure was solved in the space group C2 (Table 1) by molecular replacement with PHASER[84] using a previously solved Fab7D6 structure as the search probe. After rigid-body refinement, clear helical density could be seen in the mFo-DFc electron density maps. Using Phenix.refine[85] simulated annealing refinement, it became clear that the ApoL1-NTD had formed a domain-swapped dimer with an adjacent symmetry molecule. The final refined ApoL1-NTD model was missing residues 61–88, which was confirmed by mass spectrometry analysis of crystals. The final model contained 97.30% of the residues within favored regions of the Ramachandran plot and 0.19% of outliers. Final Clashscore: 5.35.

Crystals of the complex of Fab7D6 with the BH3-like peptide E152–H169 (synthesized by NEO Scientific, Cambridge, MA, USA) were grown in 1.8M ammonium citrate pH 7.0 and were preserved by adding 30% glycerol to the reservoir and suddenly immersed in liquid nitrogen. Data were collected at ALS 5.0.2 and 100 K/−173.5 °C at a wavelength of 1 Å and processed to 2.38 Å. The structure was solved in the space group P3₁21 by molecular replacement using PHASER[84] using a subset of the previously solved Fab7D6:ApoL1-NTD structure. The final refinement was performed in Phenix.refine[85] with TLS restraints applied. Final refinement used anisotropically scaled data from Staraniso[87] to 2.16 Å and Phenix.refine (Table 1). The final model contained 98% of the residues within favored regions of the Ramachandran plot and 0% of outliers. Final Clashscore: 4.04.

All structural figures were made by using the PyMOL Molecular Graphics System, Version 2.0 (Schrödinger, LLC).

### Attempts at crystallizing full-length ApoL1 and truncated ApoL1 forms encompassing the H-L-H transmembrane region

Additional experiments were carried out to determine whether longer constructs of ApoL1 could be crystallized. However, these attempts were unsuccessful, as we were unable to obtain well-diffracting crystals. The constructs were based on ApoL1, RefSeq "G4" (NM_003661, with E150/M228/R255). Expression was carried out using either baculovirus or E. coli constructs using an N-terminal His-tag.

For insect cell expression, the baculovirus constructs E28–S325, E28–E260, and the full-length E28–L398 were expressed in Sf9 cells with baculovirus media in a 4l Wave® bioreactor for 72 h. The baculovirus cell pellet was thawed into 0.25M NaCl, 50mM HEPES pH 7.5 containing 0.025% n-Dodecyl-β-D-Maltopyranoside (DDM). The mixture was homogenized and lysed using a Microfluidizer (Microfluidics Corp.). The suspension was clarified by ultracentrifugation. The

supernatant was loaded onto a nickel resin column (GE, 5 ml) equilibrated with 0.25M NaCl, 50mM HEPES pH 7.5, 50mM imidazole plus 0.025% DDM, and eluted with 300mM imidazole in 0.25M NaCl, 50mM HEPES pH 7.5 and 0.025% DDM. The protein peak corresponding to ApoL1 protein was eluted directly onto a S75 (16/60) size-exclusion column pre-equilibrated with 0.25M NaCl, 25mM HEPES pH 7.5 and containing one of the following detergents at their CMC: 4-Cyclohexyl-1-Butyl-β-D-Maltoside (Cymal 4), 5-Cyclohexyl-1-Pentyl-β-D-Maltoside (Cymal-5), 6-Cyclohexyl-1-hexyl-β-D-Maltoside (Cymal-6), 7-Cyclohexyl-1-Heptyl-β-D-Maltoside (Cymal-7), DDM, n-Undecyl-β-D-Maltopyranoside (UDM), n-Decyl-β-D-Maltopyranoside (DM), n-Nonyl-β-D-Maltopyranoside (NM), n-Decyl−β−D−Glucopyranoside (βDG), n-Nonyl−β−D−Glucopyranoside (βNG), n-octyl-β-D-glucopyranoside (βOG), n-Decyl-B-D-Thioglucopyranoside (βTG), N-Methyl-N-(1-Oxododecyl)-Glycine, Sodium Salt (Sarkosyl), n-Tetradecyl-N,N-Dimethyl-3-Ammonio-1-Propanesulfonate • Dimethyl(3-sulfopropyl)Tetradecyl-Ammonium Hydroxide, Inner Salt (Anzergent 3-14) and (3α-hydroxy-7α,12α-di-((O-ß-D-maltosyl)-2-hydroxyethoxy)-cholane (Façade-EM). For each ApoL1 construct and for each condition, the elution protein peak corresponding to ApoL1 was pooled and concentrated to 10 mg/ml and used for crystallization trials either as apo-form or in complex with Fab6D12, Fab7D6, or Fab3B6.

For E. coli expression, the ApoL1 constructs E28–S325, E28–E260, and E28–L398 were transformed with pNIC28_APOL1, expressed in E. coli BL21 (DE3)-RIL cells (Agilent Technologies) and inoculated the following day into 100ml LB media overnight. Five ml of overnight inoculum was added to 500ml auto-induction medium (TB) with omission of glycerol from the medium (2L total). After 24 h at 37 °C, with rigorous shaking, cells were lysed using high pressure (Microfluidizer–Microfluidics) in lysis buffer containing 50mM Tris-HCl, pH 8.5, 5mM EDTA, 0.5mM DTT, 0.5mM PMSF. Full-length ApoL1 and the two truncated forms were pelleted as insoluble material at 26,000 × g and washed with lysis buffer containing 0.5M NaCl. With constant stirring at room temperature, the proteins were solubilized in 250 ml of 1% DDM, 0.25M NaCl, 50mM Tris pH 8.0 by adjusting the pH to 12 with NaOH for 2 min; the pH was then adjusted to 8.0 by titration with 1M Tris-HCl pH 7.5. The solubilized protein was cleared by centrifugation (26,000 × g) and applied to a nickel column (HisTrap, GE Life Sciences) equilibrated in binding buffer (50mM TrisHCl pH 8.0, 250mM NaCl, 0.05% w/v DDM). The ApoL1 proteins were eluted with 300mM imidazole in binding buffer and further purified (>95%) on a Superdex-200 16/60 size-exclusion column (GE Life Sciences), equilibrated with 50mM Tris-HCl, pH 8.0, 250mM NaCl and containing one of the following detergents at their CMC: (Cymal 4, 5, 6, 7, DDM, UDM, DM, NM, βDG, βNG, βOG, βTG, Sarkosyl, Anzergent 3-14 and Façade-EM (same detergent panel as for SF9-expressed ApoL1 proteins). For each ApoL1 construct and for each condition, the elution protein peak corresponding to ApoL1 was pooled and concentrated to 10 mg/ml and used for crystallization trials either as apo-form or in complex with Fab6D12, Fab7D6, and Fab3B6. Crystallization trials were carried out using the commercial high-throughput screens Memgold, Memgold 2, Memfrac, PEGRx, JCSG + suite, Crystal screen 1 and 2 (Qiagen) using a Mosquito liquid handler.

In addition, we also explored conditions that included lipids in the crystallization matrix, such lipid cubic phase (LCP) and bicelles. For the LCP experiments, the full-length ApoL1 (E28–L398) was purified in DDM as described above and then concentrated to 30 mg/ml using a spin concentrator (100kDa MW cutoff). Several LCP experiments were carried out using MAG (monoacylglycerol) 7.9, 8.9, and 9.9 as the lipid in a 2:3 protein:lipid ratio[90]. MAG 7.9 proved too viscous and MAG 9.9 (monoolein) was the optimal lipid. The Mosquito LCP liquid handler was used to set up drops against commercial high-throughput screens at 18 °C. The LCP method failed to yield any protein crystals. For the bicelle experiments, the full-length ApoL1 (E28–L398) was purified in DDM as described above and then concentrated to 20 mg/ml using a spin concentrator (100kDa MW cutoff). The following bicelle preparation protocol was used: a detergent stock of 2.5% (w/v) CHAPSO (3-((3-cholamidopropyldimethylammonio)-2-hydroxy-1-propanesulfonate) was made up in 10mM Tris pH 8.0, 100mM NaCl. To produce a 1ml bicelle stock, 0.075 g of DMPC (dimyristoylphosphatidylcholine) was weighed into a 1.5ml Eppendorf tube and 1 ml of the CHAPSO (2.5%) stock was added. The solution was vortexed, centrifuged, and then put into an ice-cold water bath and sonicated at 1-min intervals until the lipid was dissolved. The bicelle stock was then warmed to room temperature and cooled at 4 °C degrees repeatedly and mixed, until the solution appeared homogeneous and was clear upon warming. The bicelle stock (7.5% DMPC, 2.5% CHAPSO, 10mM Tris pH 8.0, 100mM NaCl) in its liquid phase was then mixed with ApoL1 protein at a ratio of 5:1 and incubated overnight at 4 °C. Commercial high-throughput screens were set up with the Mosquito liquid handler using 0.1μl complex mixture and 0.1μl reservoir at 18 °C. Hexagonal crystal rods formed from PEGRx (0.2% Tasimate pH 7.0, 0.5% 2-propanol, 0.1M imidazole pH 7.0, 8% PEG 3350) but failed to diffract.

### NMR data collection

For pH studies, ¹⁵N-ApoL1-NTD NMR samples were prepared after purification at 200 μM in 25mM Tris pH 7.0, 150mM NaCl (pH 7.0), or buffer exchanged into 25mM sodium phosphate pH 6.0, 150mM NaCl (pH 6.0); 25mM MES pH 5.5, 150mM NaCl (pH 5.5); or 25mM sodium acetate pH 5.0, 150mM NaCl (pH 5.0). All NMR samples were supplemented with 5% D₂O. For each sample, an identical ¹⁵N-HSQC spectrum was collected at 37 °C on a 600-

MHz Bruker Spectrometer with a cryogenically cooled probe. Spectra were referenced directly ($^1$H) or indirectly ($^{15}$N) to an internal DSS (22-dimethyl-2-sila-pentane-5-sulfonate) standard.

**NMR assignments and structure determination.** A $^{13}$C$^{15}$N-ApoL1-NTD NMR sample was prepared at a concentration of 0.9 mM in 25 mM MES, 25 mM NaCl, 1 mM EDTA, pH 5.5 with 5% D$_2$O. Data were collected at 37 °C on an 800-MHz Bruker Spectrometer with a cryogenically cooled probe or a 600-MHz Bruker Spectrometer with a cryogenically cooled probe. Backbone assignments were determined sequentially using $^{15}$N-HSQC, CBCA(CO)NH, HNCA, HNCACB, HNCO, HN(CA)CO, (H)CC(CO)NH, and $^{15}$N-NOESY-HSQC (150 ms mixing time) experiments. Side chain assignments were determined using $^{13}$C-HSQC (aliphatic), $^{13}$C-TROSY (aromatic), HCCH-TOCSY, and $^{13}$C-NOESY-HSQC (aliphatic and aromatic, 150 ms mixing times) experiments. Stereospecific assignments of valine and leucine methyl groups were determined by collecting $^{13}$C-HSQC and $^{13}$C-CT-HSQC experiments collected on 0.9 mM 1:9 $^{13}$C:$^{12}$C-labeled ApoL1-NTD as described by as described by Senn et al.[91]. Spectra were referenced directly ($^1$H) or indirectly ($^{13}$C and $^{15}$N) to an internal DSS standard. All peak picking and assignments were done using CcpNMR Analysis v2.4[92]. Assignment completion percentage, with backbone completion in parentheses: total: 79.7 (90.2) 1H: 83.5 (88.4), $^{13}$C: 78.5 (91.5), $^{15}$N: 67.7 (87.5).

For multiple residues in ApoL1-NTD, in addition to the major state, an additional minor state was observed; because of the relatively low intensity of these peaks, we did not characterize them and utilized only peaks associated with the major state for structure determination. Dihedral angles were estimated using TALOS+[93]. The CYANA v3.97 NOE assignment and structure determination package were used to determine NOE assignments and complete the initial structure calculation[94,95]. Sum of $r^{-6}$ averaging was used for all NOEs. During each round of refinement, 100 structures were generated, with the 20 lowest target function structures proceeding to the next round. After the final round, 100 structures were calculated and subsequently refined in explicit water using the PARAM19 force field in CNS v1.2[96,97] and the WaterRefCNS package developed by Dr Robert Tejero, with the 20 lowest energy structures presented here. Structures were evaluated using PROCHECK-NMR, with statistics presented in Table 2. Ramachandran statistics: most favored: 96.0%, additionally allowed: 4.0%, generally allowed: 0.0%, disallowed: 0.0%.

The helix boundaries of the NMR solution structure of ApoL1-NTD, as well as the crystal structures of ApoL1-NTD and ApoL2-NTD, were determined by using CCP4[98] and ProCheck[99].

**Structure prediction and analysis.** Structural models of the NTD of ApoL3 and ApoL4 were generated using comparative modeling on the Rosetta webserver[72] (https://robetta.bakerlab.org/). The high sequence similarity, conservation of secondary structure elements and helix boundaries between ApoL1, ApoL2, ApoL3, and ApoL4 was verified by sequence alignment (ClustalO Alignment)[100] and secondary structure prediction (JPred)[71]. Sequence alignment was visualized and analyzed using JalView[101]. To generate models of the NTD of ApoL3 and ApoL4, we used the NMR structure of the ApoL1-NTD as a template and the corresponding primary sequence of ApoL3 (F73–T132; isoform 1, NP_663615.1) and ApoL4 (F29–P88; isoform 2, NP_663693) as inputs. The generated models were superimposed and analyzed using PyMOL 2.2.3 (Schrödinger, LLC).

**Surface plasmon resonance (SPR).** The binding affinities of the three mouse anti-human ApoL1 antibodies Ab7D6, Ab6D12, and Ab3B6[61,64] were determined using a Biacore™ T200 SPR system (GE Healthcare). The experiments were carried out by first coating an anti-murine Fc antibody (Cytiva, cat# BR100838) on a BIAcore™ carboxymethylated dextran CM5 chip using the Amine Coupling Kit (GE Healthcare). The anti-ApoL1 antibodies were then captured on CM5 biosensor chips to achieve ~250 response units. Fivefold serial dilutions of ApoL1 in the range of 0.8–500 nM in HBS-P buffer (10 mM HEPES pH 7.4, 150 mM NaCl, 0.005% surfactant P20) were injected at 37 °C with a flow rate of 100 μl/min. Association rates ($k_{on}$) and dissociation rates ($k_{off}$) were calculated using a 1:1 Langmuir binding model (Biacore T200 evaluation software version 3.0). The equilibrium dissociation constant ($K_D$) was calculated as the ratio $k_{off}/k_{on}$ and reported as the average ± S.D. of three independent experiments.

For experiments with Bcl-2 pro-survival proteins, measurements were performed on a Biacore™ S200 instrument (GE Healthcare) using HBS-EP (100 mM HEPES pH 7.4, 150 mM NaCl, 3 mM EDTA, 0.005% surfactant P20) as running buffer at 25 °C. Generally, Bcl-2 proteins were immobilized on a CM5 sensor chip directly via the Amine Coupling Kit, or indirectly using the His-Capture Kit (GE Healthcare). Human recombinant Bcl-2 proteins were purchased from R&D Systems: Bcl-w, catalog # 824-BW-050; Bcl-x$_L$, catalog # 894-BX-050; Bfl-1, catalog # 1160-A1-050; Bcl-2, catalog # 827-BC-050). His-tagged human Mcl-1 was purchased from Novus Bio (catalog # NBP2-51510). The following 25-mer peptides were synthesized by ABclonal (>95% purity):

ApoL1-BH3 [146]RLKSELEDNIRRLRALADGVQKVHK[170],
ApoL1-BH3 (I155A) [146]RLKSELEDNARRLRALADGVQKVHK[170],
ApoL2-BH3 [87]RLKRELEDHIRKLRALAEEVEQVHR[111],
Bid [77]ESQEDIIRNIARHLAQVGDSMDRSI[101],

Bid (A87I) [77]ESQEDIIRNIIRHLAQVGDSMDRSI[101],
Bim [141]DMRPEIWIAQELRRIGDEFNAYYAR[165],
Bim (A149A) [141]DMRPEIWIIQELRRIGDEFNAYYAR[165].

The peptides were dissolved in DMSO as 10 mM stocks. A threefold dilution series of ApoL1, ApoL2, Bid, or Bim peptides and of purified flag-tagged ApoL1 and flag-tagged ApoL2 proteins was prepared in running buffer. Final concentrations for binding analytes were at least ten times the $K_D$ for the Bim and Bid peptides. Absence of binding of the ApoL1-BH3 and ApoL2-BH3 peptides and proteins was initially probed at 100 nM and re-probed at 3 μM final concentration to verify absence of interaction with the Bcl-2 family members. All kinetic measurements were performed using single-cycle kinetics and referenced by subtracting the signal to the blank flow cell. Referenced datasets that showed binding were fitted to a 1:1 Langmuir binding model using the GE Biacore™ S200 instruments software (GE Healthcare). Final kinetics and standard deviations were calculated based on at least three independent experiments.

**Immunoprecipitation and immunofluorescence of ApoL1-6.** COS7 cells (African green monkey kidney) were maintained in high-glucose DMEM with 10% FBS (VWR/Seradigm, catalog # 97068-077), 1% nonessential amino acids (VWR/Hyclone, catalog # 16777-186), and 2 mM L-glutamine (Genentech). For immunoprecipitations, transient transfection of 10 μg human untagged ApoL1, ApoL2, or C-terminally tagged ApoL3-Myc-Flag, ApoL4-Myc (see Scales et al.[64] for details) or ApoL5-Myc-Flag (NM_030642.1 synthesized by GenScript) in 15-cm dishes was performed using Xtreme HP Transfection Reagent (Roche). Due to difficulties in transfecting ApoL6-myc-FLAG[64], a stable cell line was made using a doxycycline-inducible PiggyBac vector, as previously described[61], which was selected with 4 μg/ml puromycin and induced for 48 h using 0.5 μg/ml doxycycline for immunoprecipitation experiments. For immunofluorescence $0.9 \times 10^4$ COS7 cells/well were plated in 8 well LabTekII slides for 24 h, then transiently transfected with 0.25 μg DNAs (including ApoL6-myc-FLAG) and 0.7 μl Fugene HD (Promega, catalog # E231A) for 48 h.

For immunofluorescence experiments, transiently transfected COS7 cells were fixed for 20 min at room temperature in 3% paraformaldehyde (Electron Microscopy Sciences) in PBS, quenched for 10 min with 50 mM NH$_4$Cl in PBS, then permeabilized with 0.1% Triton-X-100 in PBS for 4 min. After blocking for 20 min in PBS containing 10% FBS and 5% BSA, mouse anti-ApoL1 antibodies Ab7D6, Ab6D12, and Ab3B6 were applied at 2 μg/ml in PBS for 1 h at room temperature, followed by 1:800 Alexa488-anti-mouse (Jackson ImmunoResearch, catalog # 715-546-150) for 1 h and then mounted in ProlongGold with DAPI (Thermo Fisher, catalog # P36931). ApoLs were co-stained with 1:200 anti-myc tag rabmab 71D10 (Cell Signaling, catalog #2278S) and 1:800 Dy649-anti-rabbit (Jackson ImmunoResearch, catalog # 711-496-152) to verify transfection (or anti-myc specificity in the case of untagged ApoL1 and ApoL2), as well as to confirm colocalization of any anti-ApoL signals. Images were acquired at 63× (N.A. 1.4) on a Zeiss AxioImager M2 with a PhotoMetrics HQ2 camera, and processed in Adobe Photoshop CC2019.

For immunoprecipitation experiments, cells were lysed for 30 min at 4 °C in 1 ml of M-PER Mammalian Protein Extraction Reagent (Thermo Scientific, catalog # 78503), a gentle non-denaturing lysis buffer, complemented with protease inhibitors (1 mM PMSF and cOmplete™ mini, Roche, catalog # 11836153001) and proteins were quantified using the Pierce BCA Protein Assay kit (catalog #23225). Lysate corresponding to 35 μg of total protein was used to immunoprecipitate ApoL1-3, and 500 μg total protein for ApoL4-6 due to poor expression of the latter. Ab7D6, Ab6D12, and Ab3B6 (20 μg each) or mouse IgG (Zenon labeling kit component B) (Thermofisher, catalog # Z25002) were pre-immobilized on magnetic beads (Dynabeads Protein G, Thermo Fisher Scientific, catalog #10004D) and incubated with cell lysates for 20 min at RT on a rotor, followed by four washes with PBS/0.02% Tween-20 and elution in 30 μl of glycine pH 2.5. The eluate was neutralized with 1/10th volume 1M Tris pH 8.0, 4x Nupage™ LDS Sample Buffer (Thermo Fisher Scientific, catalog # NP0007) and 10x NuPage™ Sample Reducing Agent (Thermo Fisher Scientific, catalog # NP0004). Samples were run on 4–12% Bis-Tris gels (NuPage™, catalog # NP0321) in 1x MOPS running buffer (Thermo Fisher Scientific, catalog #NP0001), then transferred onto nitrocellulose membranes (iBlot2, Thermo Fisher Scientific, catalog # IB23001). Immunoprecipitated ApoL1 and ApoL2 were detected with 1:1000 anti-ApoL1/2 rabbit polyclonal (Proteintech, catalog # 11486-2-AP) antibody, while Myc-tagged ApoL3-6 were detected with 1:1000 rabbit monoclonal anti-Myc tag 71D10, each followed by 1:10,000 HRP-anti-rabbit IgG (Jackson, catalog # 711-036-152). Secondary antibodies were detected using enhanced chemiluminescence (ECL Prime; General Electric Healthcare).

**Size-exclusion chromatography coupled with multi-angle laser light scattering (SEC-MALS).** The Fab7D6:ApoL1-NTD complex was formed by adding a twofold molar excess of ApoL1-NTD (46.8 μM) with Fab7D6 in PBS buffer in a total volume of 70 μl and incubating for 30 min at room temperature. The Fab7D6:ApoL1-NTD complex or the individual components Fab7D6 and ApoL1-NTD (50–80 μg) were applied to the column (XBridge BEH 200Å/Waters) of an Agilent chromatography system (1260 Infinity/Agilent Technology) connected to a Wyatt DAWN HELEOS II and an Optilab T-rEX detector. The proteins were eluted in PBS, 0.02% sodium azide at a flow rate 1 ml/min. The data were analyzed using the

ASTRA 7.2 software (Wyatt Technology). The determined molecular masses were the average ± SD of three independent experiments.

**Biolayer interferometry.** An Octet Red System (PALL ForteBio) was used to measure the binding of the murine Ab6D12[61,64] to full-length ApoL1 (ApoL1) and its mutant form in which the R150 residue was replaced with an alanine (ApoL1-R150A). ApoL1-R150A was constructed by DNA synthesis using the ApoL1 sequence (RefSeq "G4," NM_003661, with E150/M228/R255) with an N-terminal flag-TEV cleavage sequence tag as the template (GenScript Biotech). The protein was expressed in SF9 insect cells and purified as described for ApoL1[61]. Anti-mouse-Fc capture biosensors (AMC-PALL-ForteBio) were hydrated in assay buffer (50mM Tris, 300mM NaCl, 1mg/ml BSA, 0.1% Tween-20). The biosensors were transferred into wells of 384-tilted-bottom microplates (Fortebio Cat#18-0019) containing assay buffer for baseline determination and then placed into wells containing 6.25 μg/ml of Ab6D12 in assay buffer. Subsequently, the biosensors were transferred to wells containing assay buffer to remove unbound Ab6D12. In the association step, the biosensors with bound Ab6D12 were placed into wells containing 50 nM of ApoL1 or ApoL1-R105A, which was followed by the dissociation step in wells containing assay buffer. Data analysis was performed with the Octet software.

**Statistics and reproducibility.** Standard deviations were calculated based on at least three independently performed experiments (Table 3 and Supplementary Table 1).

**Reporting summary.** Further information on research design is available in the Nature Research Reporting Summary linked to this article.

## Data availability

The X-ray structures of Fab6D12:ApoL1-NTD (PDB ID: 7LF7), Fab6D12:ApoL2-NTD (PDB ID: 7LF8), Fab3B6:ApoL1-NTD (PDB ID: 7LFA), Fab7D6:ApoL1-NTD (PDB ID: 7LFB), and Fab7D6:ApoL1-peptide (PDB ID: 7LFD), as well as the NMR structure of ApoL1-NTD (PDB ID: 7L6K) have been deposited in the Protein Data Bank. The authors declare that the data supporting the findings of this study are available within the paper and its supplementary information files. All relevant data are included in the manuscript. Reagents are available from D.K. under a material transfer agreement with Genentech Inc.

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

## Acknowledgements

We thank Terry Lipari for ApoL1 purifications and antibody binding assays. Synchrotrons at the ALS and the APS are supported by the Director, Office of Science, Office of Basic Energy Sciences (BES), of the U.S. Department of Energy (DOE) under contracts DE-AC02-05CH11231 and DE-AC02-06CH11357, respectively. The Berkeley Center for Structural Biology is supported in part by the NIH, NIGMS, and the HHMI. Supporting institutions of the Southeast Regional Collaborative Access Team (SER-CAT) 22-ID beamline at APS may be found at www.ser-cat.org/members.html. The SSRL Structural Molecular Biology Program is supported by the DOE Office of Biological and Environmental Research, and by the NIH, NIGMS (including P41GM103393).

## Author contributions

M.U. and C.E. purified and crystallized proteins, solved structures, analyzed and interpreted results, and wrote the manuscript. M.J.H. carried out NMR studies, solved the NMR structure, analyzed and interpreted results, and wrote the manuscript. S.G. performed SPR studies, Rosetta models, analyzed and interpreted results, and wrote the manuscript. P.M. designed ApoL1 and ApoL2 constructs, performed purifications, SEC-MALS and BLI studies, and analyzed and interpreted results. S.J.S. performed immunofluorescence experiments, analyzed and interpreted results, and wrote the manuscript. N.G. generated ApoL expression constructs for immunoblot studies, performed ApoL1 activity assays, and provided anti-ApoL1 antibodies. F.O. performed immunoprecipitation experiments and analyzed and interpreted the results. C.C. performed SPR studies and analyzed and interpreted the results. W.F. supervised NMR experiments, analyzed and interpreted data, and wrote the manuscript. D.K. designed and coordinated the overall study and wrote the manuscript.

## Competing interests

The authors declare no competing interests.
