## [Peer Review File · Communications Biology]

Reviewers' comments:

Reviewer #1 (Remarks to the Author):

This study represents an important step forward in the structural analysis of APOL1 and APOL family proteins. A novel structural fold was convincingly shown by independent crystal and NMR structures, confirming that the homology model based on the structure of colicin A (ref 37) is incorrect. No major revisions are necessary although the following minor points should be addressed.

Introduction lines 64-71

It should be acknowledged in the introduction that the old 2005 APOL1 domain structure (ref 37) is already disputed at the time of writing. For example, evidence that the actual pore-lining region of the APOL1 channel exists within the C-terminal domain and not within the so called "pore-forming domain" is significant (ref 41). Although already stated in the discussion section, this should also be stated in the Introduction.

Results

Line 102-103: "Full-length ApoL1 solubilization and purification required the presence of detergent and resisted attempts to form well-diffracting crystals"

Given that APOL1 channel forming and trypanolytic function likely depends on all domains of the protein it would be helpful to the field to include details of the crystallization studies employed, as follows

- 1) What is defined as "full-length"? (is it 61-398, as in ref 58?)
- 2) Whether the full length construct was tagged with FLAG, or His (not clear from reference 58) and confirm at which terminus
- 3) Which detergents were used to attempt crystallization?
- 4) Were multiple detergents tried?

Line 109: "Additional constructs that extended beyond the T172 residue were not soluble without detergent (data not shown)"

Again, given the importance of the Helix-Loop-Helix to trypanosome lysis and channel formation, it would be helpful if more information were provided here. For example, were attempts made to crystallize H-L-H containing constructs in the presence of detergent? If so, which detergents were used?

Figure 4A: Please clarify which pH was used in panel A.

Figure 4E: The labeling and color schemes are confusing. What is the basis for yellow annotation? The yellow residues are described as hydrophobic yet only the underlined ones appear so. S149 and K148 are not referred to in the legend. Additional side chains are drawn that are not referred to or labelled.

Reviewer #2 (Remarks to the Author):

The domain organization of APOL1 includes a signal peptide (aa. 1-27), pore forming domain-PFD (aa. 60-235), membrane addressing domain- MAD (aa. 238-304) and the C-terminal SRA-interacting domain (aa. 339-398). Within the PFD is a BH3 like domain (aa. 158-166). The kidney disease associated G1 and G2 variants are located in the C-terminal domain of the protein. APOL1 is known to interact with membranes, oligomerizes and forms a nonspecific cation channel. The membrane interacting regions, oligomerization patterns and essential domains required for oligomerization is unclear.

In this manuscript, the authors- purified residues D61-T172 (APOL1-NTD) without detergents using insect cell expression. This region did not oligomerize based on the SEC-MALS data. This region was crystallized in presence of three antibody chaperones Ab6D12, Ab3B6 and Ab7D6. These antibodies were previously characterized by the authors. For Ab6D12- No electron densities were observed for D61-S64 and K142-T172 and the rest of the protein formed an amphipathic 4-helix bundle. For Ab3B6 electron densities were observed until V168 and the long helix 4 in the Ab6D12 structure is bent (the antibody binding epitopes are located in this elbow). For Ab7D6, a

domain swapped dimer (1 Ab7D6 bound to each protomer which spanned N91-K170) was seen. Hence helix 1 and 2 were absent in this structure. The structure of BH3 domain was resolved in this case and the epitope binding region included the BH3 like domain. NMR was carried out at pH 5.5 as the authors observed higher peak numbers and intensities at lower pH compared to pH7.0. The structure observed is similar to the 4-helix bundle seen in the crystallographic models but with the "helix 5" in a bound state with the groove formed by the rest of the 4-helix bundle. The authors suggest that the "open state" is seen with the Ab3B6 structure. In the "open state" the LZD in the C-terminus can bind the NTD region. This model is similar to that reported for many Bcl family protein structures. The authors show that the BH3 like region of APOL1 does not bind to other bcl proteins and further show that the NTD of APOL1 structure is similar in closely related APOL2.

Experimental structural data of APOL1 structure has not been published so far, and structural information will be critical in advancing the field forward in terms of understanding kidney disease pathophysiology and developing therapeutic strategies. However, structural studies on the C-terminal region of ApoL11, incl. a recent review should be cited.

But the concerns are-

1. The manuscript presents structural data about the N-terminal domain of APOL1. Even not considering the deficiencies I mentioned below, the study does not advance our understanding of the physiological function of APOL1 or give any information about the functional consequences of APOL1 kidney disease-associated variants.

2. Regarding the crystallographic studies-

Region T173-W235 of the PFD is excluded in crystallographic studies. This excluded region contains the two predicted transmembrane TM domains in the PFD of APOL1. The authors mention that the construct which included this region was not soluble without detergents but is the region of PFD which has most functional significance (cellular localization, function as ion channel, cytotoxicity etc). The effect of the absence of this membrane associated region on the NTD is a concern.

The restraints induced by the antibody binding to APOL1 is a concern when interpreting the crystallographic structures presented. The agreement between the structures seen with the 3 antibodies are not good. For example- The helix 4 is bent in Ab3B6 structure compared to Ab6D12 structure. Can this be an artifact secondary to the antibody binding epitopes that are located in the region of helix 4 that is bent?

3. Regarding the NMR-

Data is not well presented. Data about completed assignments, missing assignments are not shown. The NMR assignments need to be deposited to BMRB and the restraint file to the PDB. Contrary to the statement in the text that the spectra are closely similar at pH5 and 6, there are several peaks which show titration behavior. The assignments, with help of the structure, should be used to identify the residues which titrate and the resonances which appear in the spectrum. A suggestion why the protein may oligomerize at higher pH through these regions should be given.

4. The "conformational freedom" of helix-5 as the authors say is not substantiated by the data presented. I don't think that the comparison of crystallographic structure with an "open state" and NMR structure showing a "closed state" is sufficient. This region did not show significant line broadening as per author's description- but assignments are not shown. NMR relaxation studies are needed to confirm this.

5. The author's argument that helix-5 exists in an open state and can bind the C-terminal LZD should be tested experimentally.

Reviewer #3 (Remarks to the Author):

SUMMARY:

APOL1 forms ion channels in lipid bilayers, lyses African trypanosomes, and is implicated in chronic kidney disease. Despite a rapid expansion of the L1 field in the past ten years, there remains a poor understanding of how the ion channel is formed. This paper makes the first successful effort to interrogate the function of this protein from the perspective of classical structural biology. Past efforts relied exclusively on *in silico* modeling; work that has been called into question over the years due to the poor homology and lack of functional similarity to the available templates. This is largely due to the unique nature of the APOL1 protein and its family members. Here, the authors have been able to generate structural data that offers a significant advancement to the field, which is particularly important in that it finally puts to bed the idea that APOL1 bears structural homology to bacterial colicins by defining the majority of the “PFD” as a four-helix bundle that diverges from the colicin counterpart (the agreement on the core fold between different crystal structures and the independent technique of NMR lends confidence). This domain of the protein was previously implicated as the functional pore of the protein, while this data suggests an alternative hypothesis.

Overall, this paper is suitable for publication with only modest revisions. It is a very nice piece of work with excellent visuals, the structural biology limited only by the highly hydrophobic nature of the channel-forming region and thus the amount of the protein that is “crystallizable.” While this is a significant limitation of the manuscript’s biological impact, the insights provided in this excellent work are still critical to moving the field forward from many likely incorrect hypotheses.

The authors’ conclusions are well supported by the generated data, and additional experiments are not required for publication. A few experiments are proposed that could increase the impact of this publication, however. More importantly, the manuscript could benefit from additional discussion expanding the interpretation of these interesting data (as detailed below).

MAJOR POINTS:

To that last point, the authors could expand the discussion section to address a key question: What does this data mean with respect to the way that APOL1 forms membrane channels? They make one very strong statement (“there is no apparent structural basis to deduce a pore-forming function of the ApoL1-NTD based on the pore-forming mechanism of these proteins.”) and subsequently briefly discuss the C-terminal pore hypothesis from the Raper/Thomson group. However, they then drop the discussion of ion channels entirely.

The answer to the question above at this point in the manuscript’s development is that “these data show that the ‘pore forming domain’ is probably not the pore.” While that is indeed a fair interpretation of the data, expanding on this discussion by generating (and discussing) two things would add to the paper and thereby the APOL1 field: (1) a more up-to-date domain map of the APOL1 protein that encompasses everything discerned by this manuscript in addition to the rest of the available data generated by the field – this one is particularly critical given that the previous model being addressed/refuted by this manuscript is some 15 years old – and (2) a more conceptual illustration of how these data may be biologically represented in the contexts of HDL biology and membrane bilayers. In Fig 4D, for example, the authors illustrate the protein as if the H-L-H’s insertion into the bilayer is facilitated by or associated with only the “open state” confirmation of the NTD. Can they discuss why? How does this data inform our understanding of channel formation?

MINOR POINTS:

(1) Lines 51-54 → The authors, as many in the APOL1 field do, have referenced some outdated data here. It remains true that the G1 protein can kill lab isolates of Tbb in vitro and protect mice from lab strain infection, however the 2017 eLife paper from the MacLeod group (PMID: 28537557) revealed that in humans, the G1's role is predominantly associated with reducing *gambiense* trypanosomiasis disease severity. They found no protective effect provided by G1 in the case of *rhodesiense* sleeping sickness, however. These data might be updated eventually with additional field studies but as of yet, this is the most up-to-date data for how the G1 and G2 "work" in humans and the L1 field never seems to mention it.

(2) Line 68 → The authors refer to SRA as a VSG. This is incorrect. SRA was derived through duplication and modification of a VSG allele but referring to it as a VSG itself would be analogous to referring to any gene duplication product as its ancestor. Furthermore, the distinct functions (and structures) of VSGs relative to SRA make such a classification of SRA as a VSG problematic.

(3) Intro general → The original alignment that suggested L1 might function analogously to a colicin was dubious at best. However, almost 20 years later, the "PFD" and "MAD," along with the already fully de-classified BH3 domain, are still mentioned approx. 10 times combined in the Introduction and Abstract. The authors are free to discuss the literature as long as they make reference to the papers that have already addressed the above, which they have done with more minor representation. But it would strengthen the manuscript if the authors were to be a little more critical of those outdated data in the introduction. It currently reads as though the PFD/MAD/BH3 combo of domain structure is still the "state-of-the-art" in the field, which is not really the case, or is *at least* not the consensus. As mentioned in the "major point," someone needs to redefine the domain structure of L1, and this paper does a pretty good job of that without actually writing that down or making a point of it. A simple fix would be to at least add the word "putative" in front of each mention of the domains in the manuscript, along with addressing my "major point." However, the authors could go much further and explicitly establish a new model/paradigm for this in the field.

(4) Line 254-256 → The authors could consider co-crystallization of the NTD with a C-terminal peptide incorporating the leucine zipper. It would also be prudent to simply assess binding of the two by SPR, as Uzureau et al. have done, albeit with less finesse than that which could be applied in this instance.

(5) Figure 6 provides additional evidence that the L1 BH3-like motif does not have any BH3-like function. This builds on data generated by a number of groups. The binding data is very convincing, but the authors could consider the following experiment: Is it possible to replace the full BH3-like motif from L1 with that of Bid? It would be intriguing to see if a truly functional BH3 domain could be transplanted into APOL1 (ensuring that the protein still functions normally by trypanolysis) and achieve binding to Bcl-X in their system. Is it something else about the L1 BH3-like motif that prevents binding? Or is it something else entirely that is elsewhere in the L1 protein that prevents binding?

(6) Line 420-422 → In ref 72, the authors could create non-functional APOL1s by either deleting the entire helix or by substituting multiple acidic residues in place of hydrophobic ones. However, more conservative mutations (i.e., trimming the critical Asp to an alanine) did not affect function

at all. Ultsch et al., likely understood that that chopping out the entire helix is going to affect protein structure/function in a way that is not easily interpretable. This issue could be considered important because many in the L1 field are still devoting resources to this BH3 story. One recommendation would be for the authors speak even more strongly about their results here and elsewhere. They and others have convincingly shown that the L1 field likely should stop calling this a BH3 domain/motif altogether (it is “just a helix”).

(7) Comparisons with Colicins/Bcl-x: Because of the calcified ideas about ApoL1 similarity to these families of proteins, more clarity here might be useful. To begin, it is a bit misleading to list H1 as “aligning” with the helices in colicin-A-PFD and Bcl-xL. A helix is a helix, and it is generally trivial to “align” a single helix from one structure to that in another, and emphasizing this might give the wrong impression of similarity that might not meaningfully exist. The real question here is do we have a similarity in the overall fold/topology that implies functional similarity. Unfortunately, there is no general consensus for what constitutes “similar enough” at the broad level of a protein fold in the context of biological significance (the devil is usually in the details, and many proteins share highly similar folds with very different functions). In this case, all of these structures are 4-helix bundles, and “counterpart” helices can be found to the ApoL1 helices- see marked-up example image included – note this is drawn for 2D image without the coordinates. This is not to say this alignment is correct, but used to suggest whether the structures or similar or not might not be obvious to everyone looking at the figures. The helices don’t superpose “well”, but they are close in space, and the non-superposition doesn’t mean that there isn’t conserved function (which is what the essence of this issue is about – is there some conserved function?). DALI/ PDBeFold failing to pull out similarity offers some help, but these are not infallible algorithms nor well-known outside of structural biology, and especially don’t attest to functional similarity.

I think the authors do a good job in arguing for the structure/function divergence, but the entrenched nature of the colicin “link” for many in the field might require more effort to dislodge (so, this is a minor point). It might be useful to emphasize that 4-helix bundles represent a massive family with very divergent functions centered on a common scaffold, and that unless there is something beyond the simple similarity in helical positioning (generally there in all members of the family to define the fold), there is no reason to assume functional similarity. Then the arguments for how function appears to differ in so many ways (as the authors note) might be better contextualized.

However, this might be overkill, but I do think the supplementary figure as is might obscure the divergence for some rather than illuminate it.

Comments to Reviewers:

Reviewer #1 (Remarks to the Author):

This study represents an important step forward in the structural analysis of APOL1 and APOL family proteins. A novel structural fold was convincingly shown by independent crystal and NMR structures, confirming that the homology model based on the structure of colicin A (ref 37) is incorrect. No major revisions are necessary although the following minor points should be addressed.

Introduction lines 64-71

It should be acknowledged in the introduction that the old 2005 APOL1 domain structure (ref 37) is already disputed at the time of writing. For example, evidence that the actual pore-lining region of the APOL1 channel exists within the C-terminal domain and not within the so called "pore-forming domain" is significant (ref 41). Although already stated in the discussion section, this should also be stated in the Introduction.

We have re-written this part of the introduction to correct this. In addition, we have modified other parts of the introduction in response to comments of other reviewers. The changes in the text are in red color throughout the manuscript and comprise the changes made in response to all three reviewers.

According to secondary structure predictions, ApoL1 is composed of amphipathic α -helices^{1,2} and contains three or four transmembrane domains^{19,43}. The C-terminal region, known as the SRA interacting domain (SRA-ID), contains a leucine zipper motif and interacts with the monomeric trypanosome surface glycoprotein SRA^{45,46}, but also with ApoL3^{47,48} and the vesicle-associated membrane protein 8 (VAMP8)⁴⁹. The extended N-terminal region (also known as the pore-forming domain, PFD³⁹) encompasses a putative transmembrane segment^{19,43} and was predicted to adopt a colicin A-like fold³⁹. This led to the proposition that the ApoL1 pore-forming mechanism is related to that of pore-forming colicins, diphtheria toxin and B-cell lymphoma 2 (Bcl-2) family members^{8,39}, which share structural similarities⁵⁰. However, this long-standing model may need revision in light of a recent study by Schaub et al.⁴³, which provides strong evidence that the pore-lining region is actually located in the C-terminal, rather than the N-terminal region of ApoL1.

Results

Line 102-103: "Full-length ApoL1 solubilization and purification required the presence of detergent and resisted attempts to form well-diffracting crystals"

Given that APOL1 channel forming and trypanolytic function likely depends on all domains of the protein it would be helpful to the field to include details of the crystallization studies employed, as follows

- 1) What is defined as "full-length"? (is it 61-398, as in ref 58?)
- 2) Whether the full-length construct was tagged with FLAG, or His (not clear from reference 58) and confirm at which terminus
- 3) Which detergents were used to attempt crystallization?
- 4) Were multiple detergents tried?

-Line 109: "Additional constructs that extended beyond the T172 residue were not soluble without detergent (data not shown)"

Again, given the importance of the Helix-Loop-Helix to trypanosome lysis and channel

formation, it would be helpful if more information were provided here. For example, were attempts made to crystallize H-L-H containing constructs in the presence of detergent? If so, which detergents were used?

Response to 1)-4) and Line109 item: The full-length form used for crystallization studies is ApoL1 E28-L398 with an N-terminal flag (NTF) or His (NTH) tag. This construct is different from that described in the methods as "full-length" which was D61-L398 NTF and which was originally described by Gupta et al. 2020. We apologize for the confusion. We have now clearly defined the D61-L398 NTF and for what experiments it was used (SPR and Biolayer interferometry). As requested by the reviewer, we have now added a comprehensive method section on our unsuccessful crystallization trials with the full-length construct E28-L398, as well as with other truncated ApoL1 forms encompassing the H-L-H region and disclose the detergents and lipid bicelles used. This reflects a tremendous amount of work prior to our more successful efforts on the ApoL1-NTD. We mention these attempts in the results in abbreviated form and refer to the method section for details. We hope that this is agreeable to the reviewer.

Figure 4A: Please clarify which pH was used in panel A.

The NMR structure was determined at pH 5.5 and this is now indicated in the legend.

Figure 4E: The labeling and color schemes are confusing. What is the basis for yellow annotation? The yellow residues are described as hydrophobic yet only the underlined ones appear so. S149 and K148 are not referred to in the legend. Additional side chains are drawn that are not referred to or labelled.

We have simplified this figure by using a single color (cyan) for the helix 5 and only show the side chains of residues that engage with the hydrophobic portion (L151, I155, L158) and with the acidic portion (K148, R156, R159) of the groove. We have changed the legend accordingly.

Figure 4. E. Surface representation of the four-helix core (surface is colored according to approximate net electrostatic potential: blue, positive; red, negative) with helix 5 (cyan) and the helix 4-5 linker (blue) as cartoon. The shallow groove that harbors helix 5 (mainly its BH3-like portion) is composed of a hydrophobic and a negatively charged portion. The main

hydrophobic contacts are made by helix 5 residues L151, I155 and L158 forming the LxxxIxxL motif (underlined residues), whereas K148, R156 and R159 engage with the acidic portion.

Reviewer #2 (Remarks to the Author):

The domain organization of APOL1 includes a signal peptide (aa. 1-27), pore forming domain-PFD (aa. 60-235), membrane addressing domain- MAD (aa. 238-304) and the C-terminal SRA-interacting domain (aa. 339-398). Within the PFD is a BH3 like domain (aa. 158-166). The kidney disease associated G1 and G2 variants are located in the C-terminal domain of the protein. APOL1 is known to interact with membranes, oligomerizes and forms a nonspecific cation channel. The membrane interacting regions, oligomerization patterns and essential domains required for oligomerization is unclear.

In this manuscript, the authors- purified residues D61-T172 (APOL1-NTD) without detergents using insect cell expression. This region did not oligomerize based on the SEC-MALS data. This region was crystallized in presence of three antibody chaperones Ab6D12, Ab3B6 and Ab7D6. These antibodies were previously characterized by the authors. For Ab6D12- No electron densities were observed for D61-S64 and K142-T172 and the rest of the protein formed an amphipathic 4-helix bundle. For Ab3B6 electron densities were observed until V168 and the long helix 4 in the Ab6D12 structure is bent (the antibody binding epitopes are located in this elbow). For Ab7D6, a domain swapped dimer (1 Ab7D6 bound to each protomer which spanned N91-K170) was seen. Hence helix 1 and 2 were absent in this structure. The structure of BH3 domain was resolved in this case and the epitope binding region included the BH3 like domain. NMR was carried out at pH 5.5 as the authors observed higher peak numbers and intensities at lower pH compared to pH7.0. The structure observed is similar to the 4- helix bundle see in the crystallographic models but with the "helix 5" in a bound state with the groove formed by the rest of the 4-helix bundle. The authors suggest that the "open state" is seen with the Ab3B6 structure. In the "open state" the LZD in the C-terminus can bind the NTD region. This model is similar to that reported for many Bcl family protein structures. The authors show that the BH3 like region of APOL1 does not bind to other Bcl proteins and further show that the NTD of APOL1 structure is similar in closely related APOL2.

Experimental structural data of APOL1 structure has not been published so far, and structural information will be critical in advancing the field forward in terms of understanding kidney disease pathophysiology and developing therapeutic strategies. However, structural studies on the C-terminal region of ApoL11, incl. a recent review should be cited.

We have modified portions of the introduction according to other reviewers' suggestions and have also expanded on discussing the structural work on the C-terminal region as requested. This includes the work by Pays et al. 2014 [ref#17], Madhavan et al. 2017 [Ref#49], Sharma et al. 2016 [ref#58] and the recent review by Madhavan and Buck 2021 (review) [ref#60]. The changes in the text are in red color throughout the manuscript and comprise the changes made in response to all three reviewers.

But the concerns are:

1. The manuscript presents structural data about the N-terminal domain of APOL1. Even not considering the deficiencies I mentioned below, the study does not advance our understanding of the physiological function of APOL1 or give any information about the functional consequences of APOL1 kidney disease-associated variants.

We are aware that our work on the ApoL1 N-terminal domain cannot and does not provide a structural basis to explain the disease-causing ApoL1 variants located at the C-terminus, or the physiologic function of the wildtype form; this would require a structure of the full-length ApoL1. In fact, we have carried out extensive efforts with the full-length form and truncated ApoL1 versions encompassing the H-L-H transmembrane region but without success (see point 2 below). We think it is unreasonable to expect that the first structural report on ApoL1 would answer these "big" questions. Realistically, to obtain a structure of the full-length membrane-embedded ApoL1, which might explain its ion channel function and perhaps even the impact of the disease variants will likely be a lengthy and challenging endeavor. 18 years after the discovery of ApoL1 as the lytic factor in serum (Vanhamme et al 2003; ref#45) the structural knowledge of ApoL1 is limited to molecular models of N- and C-terminal domains; therefore, we consider our work on the ApoL1-NTD as an important landmark, especially since it refutes long-held notions about its relationship to colicins and the presumed link of the BH3 region with Bcl-2 family proteins. In addition, we report that the identified novel fold is not limited to ApoL1 but extends to other ApoL family members, none of which have been structurally explored so far. We strongly believe that our findings provide an important foundation for the ApoL1 field, enabling future structural exploration of ApoL1 and other family members.

2. Regarding the crystallographic studies - Region T173-W235 of the PFD is excluded in crystallographic studies. This excluded region contains the two predicted transmembrane TM domains in the PFD of APOL1. The authors mention that the construct which included this region was not soluble without detergents but is the region of PFD which has most functional significance (cellular localization, function as ion channel, cytotoxicity etc). The effect of the absence of this membrane associated region on the NTD is a concern.

We agree that the predicted transmembrane domain in the T173-W235 region is functionally important and structural insights into an extended NTD comprising the region would be highly desirable. We have actually dedicated a huge amount of work to pursue this objective by use of various ApoL1 constructs comprising this region and dozens of crystallization trials with differing detergents, lipids and bicelles. However, this approximately two-year effort did not yield any tangible results. In response to another reviewer's suggestion, we have now added a method section with details about the nature of the constructs and crystallization conditions used, as this was considered helpful to the ApoL1 field.

However, given the well-behaved nature and stability of the isolated NTD, especially the four-helix core, we believe that it is reasonable to assume that this domain structure is representative of an ApoL1 structure that would include the transmembrane segment. The finding that the isolated NTD readily folds into a four-helix core conformation is consistent with the assumption that the transmembrane segment, which is sequestered away within the membrane compartment and, thus, not in immediate contact with the NTD, is not required to induce this fold.

The restraints induced by the antibody binding to APOL1 is a concern when interpreting the crystallographic structures presented. The agreement between the structures seen with the 3 antibodies are not good. For example- The helix 4 is bent in Ab3B6 structure compared to Ab6D12 structure. Can this be an artifact secondary to the antibody binding epitopes that are located in the region of helix 4 that is bent?

The X-ray structures of the Fab6D12 and Fab3B6 complexes are in good agreement in respect to the four-helix core (also in agreement with the NMR structure). As pointed out by the reviewer, the two structures differ in the region beyond residue M124, forming a bend in the Fab3B6 structure, but not in the Fab6D12 structure, which shows an extended helix 4. This long helix 4 is artificially stabilized by a symmetry-related molecule in the crystal lattice (Fig.1A), preventing it to adopt the bent conformation as seen in the Fab3B6 structure. Therefore, it is not an intrinsic property of Fab6D12 that would prevent the helix 4 to adopt a bent conformation, which we believe is physiologically relevant. Biolayer interferometry experiments demonstrated that 3B6 is able to bind to the preformed ApoL1-NTD:6D12 complex and conversely 6D12 can bind to the preformed ApoL1-3B6 complex (unpublished). This strongly suggests that ApoL1 can readily adopt the "bent" conformation and that the extended helix 4 observed in the 6D12 complex is a crystallization artifact. Importantly, the study by Gupta et al. 2020 (ref#61) showed that Ab3B6 recognized ApoL1 on the surface of podocytes and CHO cells, strongly indicating that the "bent" conformation (seen in the 3B6 structure) of the helix 4 represents a natural state. We conclude that the extended helix 4 in the 6D12 complex may not reflect an important natural state and have written this more clearly in the result section.

3. Regarding the NMR- Data is not well presented. Data about completed assignments, missing assignments are not shown. The NMR assignments need to be deposited to BMRB and the restraint file to the PDB. Contrary to the statement in the text that the spectra are closely similar at pH5 and 6, there are several peaks which show titration behavior. The assignments, with help of the structure, should be used to identify the residues which titrate and the resonances which appear in the spectrum. A suggestion why the protein may oligomerize at higher pH through these regions should be given.

Supplementary Table 3 details the backbone and total assignment coverage, along with structural statistics related to the solution structure determination. The assignments have been submitted to the BMRB (to be released under ID# 30832) and the structural ensemble and restraint files to the PDB (be released under ID# 7L6K). For reference, we have now included in Supplementary Figure 4 an HSQC spectrum labeled with all completed backbone amide assignments.

Most proteins exhibit notable pH-dependent chemical shift perturbations even without significant changes to the ground-state structure. These perturbations are attributable to changes in protonation state of titratable sidechains, most notably for this pH range of His and Gln. Nonetheless, the overall pattern of peaks between pH 7.0 and 5.5 remains consistent, indicating that the ground-state structure of the protein remains largely similar. We have adjusted the language of the description to more accurately describe the observed pH-dependent changes to the spectrum (see below).

Given the still-dynamic nature of the protein at pH 5.5, we expect that the poor spectral quality at pH 7.0 is more likely due to increased conformational flexibility rather than oligomerization/aggregation; the reviewer is correct that weak self-association could also contribute to the observed poor spectral quality. Given the high protein concentrations utilized for NMR, we hesitate to attribute any functional relevance to any pH-dependent self-association observed. We have adjusted the language here as well to acknowledge that self-association may also play a role here.

Notably, the spectral pattern at pH 7.0 remains similar to that observed at lower pH conditions, (Supplementary Fig. 4), suggesting that the lowest energy conformation sampled at pH 7.0 is largely

similar to that sampled at pH 5.5, but that this state is more stable and/or less prone to aggregation at the lower pH.

4. The “conformational freedom” of helix-5 as the authors say is not substantiated by the data presented. I don’t think that the comparison of crystallographic structure with an “open state” and NMR structure showing a “closed state” is sufficient. This region did not show significant line broadening as per author’s description- but assignments are not shown. NMR relaxation studies are needed to confirm this.

The ‘bound’ state observed as the dominant conformational state by NMR is structurally incompatible with the binding mode of two of the herein used antibodies, 3B6 and 7D6. Both of these antibodies bind to recombinant, detergent-solubilized full-length ApoL1 (Suppl. Table 1) and, more importantly, to podocyte and CHO cell surface-expressed ApoL1 (Gupta et al., 2020, ref#61), indicating that the ‘open’ state is accessible in the native state of the protein. Further, the antibody-bound crystal structures of 3B6 and 7D6 indicate that the NTD construct utilized for both crystallography and NMR studies here is capable of binding these antibodies as well, which requires that the NTD first samples the ‘open’ state conformation with which they are compatible. It is on these observations that we assert that the NTD must undergo conformational exchange between the ‘open’ and ‘bound’ states. Our NMR solution structure indicates that, under the conditions of the NMR experiment, the ‘bound’ state is the dominant one. The lack of notable line broadening in the spectrum at the end of helix 5 (see new Supplementary Figure 4, e.g. residues 160-165) indicates that the ‘open’ state is sampled with a low population under NMR conditions. The reviewer is correct that, if the timescale and populations are amenable, relaxation dispersion experiments may be able to elucidate the population and timescale of the exchange. However, given the additional, more populated, intermediate-to-slow conformational exchange observed in the linker region nearby, deconvoluting the multiple conformational exchange events of different timescales/populations would be a significant undertaking and well beyond the scope of this study, particularly before establishing whether the observed ‘bound’ state is functionally relevant in the context of the full-length protein.

5. The author's argument that helix-5 exists in an open state and can bind the C-terminal LZD should be tested experimentally.

We have tried crystallizing this complex (NTD-LZD) but without success. We are currently conducting NMR experiments to obtain structural details of this interaction and to test the hypothesis that the Leu zipper peptide binds to the Helix5 groove. This is a project on its own and will take time as we are using different NTD constructs and different peptides. Of course, we would also like to find out how the WT peptide affinity compares to the G2 peptide and whether their NTD binding sites are the same. These experiments are ongoing/planned. We hope that the reviewer agrees with our viewpoint that these studies are beyond the scope of the current manuscript.

Reviewer #3

SUMMARY:

APOL1 forms ion channels in lipid bilayers, lyses African trypanosomes, and is implicated in chronic kidney disease. Despite a rapid expansion of the L1 field in the past ten years, there remains a poor understanding of how the ion channel is formed. This paper makes the first successful effort to interrogate the function of this protein from the perspective of classical structural biology. Past efforts relied exclusively on *in silico* modeling; work that has been called into question over the years due to the poor homology and lack of functional similarity to the available templates. This is largely due to the unique nature of the APOL1 protein and its family members. Here, the authors have been able to generate structural data that offers a significant advancement to the field, which is particularly important in that it finally puts to bed the idea that APOL1 bears structural homology to bacterial colicins by defining the majority of the "PFD" as a four-helix bundle that diverges from the colicin counterpart (the agreement on the core fold between different crystal structures and the independent technique of NMR lends confidence). This domain of the protein was previously implicated as the functional pore of the protein, while this data suggests an alternative hypothesis.

Overall, this paper is suitable for publication with only modest revisions. It is a very nice piece of work with excellent visuals, the structural biology limited only by the highly hydrophobic nature of the channel-forming region and thus the amount of the protein that is "crystallizable." While this is a significant limitation of the manuscript's biological impact, the insights provided in this excellent work are still critical to moving the field forward from many likely incorrect hypotheses.

The authors' conclusions are well supported by the generated data, and additional experiments are not required for publication. A few experiments are proposed that could increase the impact of this publication, however. More importantly, the manuscript could benefit from additional discussion expanding the interpretation of these interesting data (as detailed below).

MAJOR POINTS:

To that last point, the authors could expand the discussion section to address a key question: What does this data mean with respect to the way that APOL1 forms membrane channels? They make one very strong statement ("there is no apparent structural basis to deduce a pore-forming function of the ApoL1-NTD based on the pore-forming mechanism of these proteins.") and subsequently briefly discuss the C-terminal pore hypothesis from the Raper/Thomson group. However, they then drop the discussion of ion channels entirely.

The answer to the question above at this point in the manuscript's development is that "these data show that the 'pore forming domain' is probably not the pore." While that is indeed a fair interpretation of the data, expanding on this discussion by generating (and discussing) two things would add to the paper and thereby the APOL1 field: (1) a more up-to-date domain map of the APOL1 protein that encompasses everything discerned by this manuscript in addition to the rest of the available data generated by the field – this one is particularly critical given that the previous model being addressed/refuted by this manuscript is some 15 years old – and (2) a more conceptual illustration of how these data may be biologically represented in the contexts of HDL biology and membrane bilayers. In Fig 4D, for example, the authors illustrate the

protein as if the H-L-H's insertion into the bilayer is facilitated by or associated with only the "open state" confirmation of the NTD. Can they discuss why? How does this data inform our understanding of channel formation?

We would like to thank the reviewer for the overall positive assessment of our work and for the insightful comments and suggestions. The changes in the text are in red color throughout the manuscript and comprise the changes made in response to all three reviewers.

(1) We have re-written large sections of the discussion and have expanded it to discuss our results in the context of the Schaub et al. JBC 2020 (ref#43) findings and state more explicitly that the "PFD", which includes the NTD, is not the pore. We further discuss the possibility that the structural stabilization of the NTD at low pH could be related to ApoL1 insertion into lipid bilayers, since the lipid insertion of the adjacent transmembrane segment seems to be favored by low pH conditions, as observed by Schaub et al. However, we are cautious not to overinterpret our results in regard to channel function and channel structure, since we really don't have any data speaking to these aspects. As outlined below (minor point 3) we are hesitant to generate a new model of ApoL1 domains for several reasons. First, this task was already accomplished by a nice cartoon in fig.8 of the recent JBC paper by Schaub et al. (ref#43). There would be nothing for us to add except that the exposed N-terminal region in their cartoon adopts a four-helix conformation. Our results don't allow us to make conclusion about other ApoL1 regions, especially since they have not been structurally examined or validated. The only other structurally explored region is the SRA-ID, which was proposed to adopt a coiled coil conformation based on molecular modeling (e.g. Sharma et al. 2016 #58, Madhavan et al. 2017 #49). This model presumes that the N-terminal helix of the coiled coil is freely available to form this structure. However, the Schaub et al. model shows that this N-terminal helix is part of the transmembrane pore and would not be available to form a coiled coil (this problem is now also mentioned in the introduction). In addition, there is no consensus on the very basic question of how many transmembrane domains there actually are (3 or 4?). The domain architecture and domain structures of ApoL1 seem to be moving targets and we prefer to stay clear from proposing yet another, potentially faulty, model at this stage, especially since our data are limited to the N-terminal portion of ApoL1 only.

(2) We have modified Fig.4D; it was not our intention to convey the impression that only the open state is associated with the H-L-H, but we realize that our cartoon may create this perception. The main point of the cartoon was to present a simplified version of the adjacent "bound" and "open" crystal and NMR structures (Fig. 4D), respectively, and convey the idea of an available groove for interaction with the Leu zipper helix of a ligand. Therefore, we have now removed the H-L-H in the cartoon and we address the membrane-anchoring of the helix 5 in the discussion section separately.

Following the reviewer's suggestion, we have added a paragraph to discuss the NTD structure in the context of HDL-bound ApoL1. While this is an intriguing question, we don't have much experimental information to validate the hypothesis that the conformational state(s) of the lipid monolayer-associated NTD in an HDL particle differs from the herein presented structures, which we believe represent the cell membrane form. The only experimental evidence comes from a large epitope mapping study by Gupta et al. 2020 (ref#61) comparing ApoL1 in podocytes and CHO cells with HDL-associated ApoL1 and we have now discussed this in an added paragraph.

MINOR POINTS:

(1) Lines 51-54: The authors, as many in the APOL1 field do, have referenced some outdated data here. It remains true that the G1 protein can kill lab isolates of Tbb in vitro and protect mice from lab strain infection, however the 2017 eLife paper from the MacLeod group (PMID: 28537557) revealed that in humans, the G1's role is predominantly associated with reducing gambiense trypanosomiasis disease severity. They found no protective effect provided by G1 in the case of rhodesiense sleeping sickness, however. These data might be updated eventually with additional field studies but as of yet, this is the most up-to-date data for how the G1 and G2 "work" in humans and the L1 field never seems to mention it.

Thank you for pointing out the eLife paper of Cooper et al. 2017 [ref#16], which we failed to mention in our submitted ms version. As remarked by the reviewer, the findings by Cooper et al. differ from in-vitro trypanosome killing experiments in that the G1 variant was not associated with protection against *T.b. rhodesiense*, but it protected against *T.b.gambiense*. We have now modified our introduction and included the Cooper et al. 2017 reference.

(2) Line 68: The authors refer to SRA as a VSG. This is incorrect. SRA was derived through duplication and modification of a VSG allele but referring to it as a VSG itself would be analogous to referring to any gene duplication product as its ancestor. Furthermore, the distinct functions (and structures) of VSGs relative to SRA make such a classification of SRA as a VSG problematic.

We thank the reviewer for this correction. Indeed, the elegant study by Zoll et al. 2018 [ref#46] on the SRA structure notes the differences between SRA and VSGs quite clearly and we should have caught this. We have now corrected our statement.

(3) Intro general: The original alignment that suggested L1 might function analogously to a colicin was dubious at best. However, almost 20 years later, the "PFD" and "MAD," along with the already fully de-classified BH3 domain, are still mentioned approx. 10 times combined in the Introduction and Abstract. The authors are free to discuss the literature as long as they make reference to the papers that have already addressed the above, which they have done with more minor representation. But it would strengthen the manuscript if the authors were to be a little more critical of those outdated data in the introduction. It currently reads as though the PFD/MAD/BH3 combo of domain structure is still the "state-of-the-art" in the field, which is not really the case, or is at least not the consensus. As mentioned in the "major point," someone needs to redefine the domain structure of L1, and this paper does a pretty good job of that without actually writing that down or making a point of it. A simple fix would be to at least add the word "putative" in front of each mention of the domains in the manuscript, along with addressing my "major point." However, the authors could go much further and explicitly establish a new model/paradigm for this in the field.

We have now avoided the terms "PFD" and "MAD" in the manuscript and only once refer to the PFD in the introduction. This should alleviate the perception that the PFD/MAD terminology is state-of-the-art. We have in fact re-written large portions of the introduction to emphasize the more current literature on the N-terminal region in respect to ion channel function and how it conflicts with the "PFD" idea. In addition, we have expanded on the SRA-ID structural models, as requested by another reviewer; in this revised version we also point out that the coiled coil model is incompatible with the pore model by Schaub et al. 2020.

(4) Line 254-256: The authors could consider co-crystallization of the NTD with a C-terminal peptide incorporating the leucine zipper. It would also be prudent to simply assess binding of the two by SPR, as Uzureau et al. have done, albeit with less finesse than that which could be applied in this instance.

We have tried crystallizing this complex but without success. We are currently conducting NMR experiments to obtain structural details of this interaction and to test the hypothesis that the Leu zipper peptide binds to the Helix5 groove. This is a project on its own and will take time as we are using different NTD constructs and different peptides. Of course, we also like to find out how the WT peptide affinity compares to the G2 peptide and whether their NTD binding sites are the same. We recently started some preliminary SPR experiments to confirm the biophysical interaction of the Leu zipper with the NTD as reported by Uzureau et al. 2020 (ref#48). However, these experiments turned out to be more challenging than expected and more optimization is needed to reduce non-specific binding interactions and various NTD and peptide constructs. We hope that the reviewer agrees with our viewpoint that these studies are beyond the scope of the current manuscript.

(5) Figure 6 provides additional evidence that the L1 BH3-like motif does not have any BH3-like function. This builds on data generated by a number of groups. The binding data is very convincing, but the authors could consider the following experiment: Is it possible to replace the full BH3-like motif from L1 with that of Bid? It would be intriguing to see if a truly functional BH3 domain could be transplanted into APOL1 (ensuring that the protein still functions normally by trypanolysis) and achieve binding to Bcl-X in their system. Is it something else about the L1 BH3-like motif that prevents binding? Or is it something else entirely that is elsewhere in the L1 protein that prevents binding?

This is an interesting idea, which we had not considered. The proposed experiments aim at understanding whether the BH3-like region would in principle be available for Bcl-2 protein interaction by use of a chimeric ApoL1 containing a grafted Bid-BH3. Based on our experiments with the 7D6 antibody we do not believe that anything in the ApoL1 protein would prevent the BH3-like region from engaging in an interaction with Bcl-2. The 7D6 epitope encompasses the BH3-like region and the 7D6 antibody binds to podocyte-expressed ApoL1, to recombinant ApoL1, to the ApoL1-NTD and to the BH3-like peptide. This indicates that in the context of all these ApoL1 constructs, BH3-like region is available to interact with the 7D6 antibody; therefore, we conclude that there is no impediment for the BH3-like region that would have prevented it from interacting with Bcl-2 proteins.

In addition, the substitution with the Bid or Bim BH3 sequence may be deleterious to trypanolysis, which would preclude the suggested control experiment. Vanwalleghem 2015 (formerly ref#72; now ref#77) demonstrated that some amino acid changes in the BH3-like region of ApoL1 were detrimental to the ability of ApoL1 to lyse trypanosomes and that trypanolysis did not require the BH3 hallmark residue Asp. Therefore, it appears that the amino acid sequence of the ApoL1 BH3-like region (but not the implied BH3 "motif") is important for eliciting proper trypanolysis. The Bid and Bim sequences are quite different from that of the ApoL1-BH3-like region and we anticipate that trypanolysis would be significantly impacted if not abolished in a chimeric construct, even if we could demonstrate that it would bind to Bcl-2 proteins by SPR. Therefore, we prefer not to engage in this line of experimentation and hope the reviewer concurs.

(6) Line 420-422: In ref 72, the authors could create non-functional APOL1s by either deleting

the entire helix or by substituting multiple acidic residues in place of hydrophobic ones. However, more conservative mutations (i.e., trimming the critical Asp to an alanine) did not affect function at all. Ultsch et al., likely understood that that chopping out the entire helix is going to affect protein structure/function in a way that is not easily interpretable. This issue could be considered important because many in the L1 field are still devoting resources to this BH3 story. One recommendation would be for the authors speak even more strongly about their results here and elsewhere. They and others have convincingly shown that the L1 field likely should stop calling this a BH3 domain/motif altogether (it is “just a helix”).

We agree that the ref#72 (now ref#77) study by Vanwalleghen et al. 2015 actually corroborates the conclusion that there is no functional BH3 motif in ApoL1, since the mutation of the hallmark Asp residue seems to have no impact on trypanolysis. The interpretation of this experiment seems straightforward (even though not discussed in the ref#72 paper), unlike some other modifications, including the deletion experiment. We have now modified this section in the discussion to emphasize that the function is unrelated to what is called the "BH3 motif" and make reference to Vanwalleghem et al. 2015.

However, the ApoL1-BH3-like region has undeniable sequence and structure similarity to true BH3-only proteins. Therefore, we decided to keep the term "BH3-like region" since the "-like" terminology has been widely used for many other protein domains/regions to convey the relatedness in sequence/structure (and apart from function). Galindo-Moreno et al. 2014 (ref#51) also used the term "BH3-like" for ApoL2 in their study showing that this region does not promote cytotoxicity, even though it is sequence-related to BH3 domains. In this discussion section we propose to use this term, rather than the commonly used "BH3 domain" or "BH3 motif" instead.

(7) Comparisons with Colicins/Bcl-x: Because of the calcified ideas about ApoL1 similarity to these families of proteins, more clarity here might be useful. To begin, it is a bit misleading to list H1 as “aligning” with the helices in colicin-A-PFD and Bcl-xL. A helix is a helix, and it is generally trivial to “align” a single helix from one structure to that in another, and emphasizing this might give the wrong impression of similarity that might not meaningfully exist. The real question here is do we have a similarity in the overall fold/topology that implies functional similarity. Unfortunately, there is no general consensus for what constitutes “similar enough” at the broad level of a protein fold in the context of biological significance (the devil is usually in the details, and many proteins share highly similar

– note this is drawn for 2D image without the coordinates. This is not to say this alignment is

correct, but used to suggest whether the structures or similar or not might not be obvious to everyone looking at the figures. The helices don't superpose "well", but they are close in space, and the non-superposition doesn't mean that there isn't conserved function (which is what the essence of this issue is about – is there some conserved function?). DALI/ PDBeFold failing to pull out similarity offers some help, but these are not infallible algorithms nor well-known outside of structural biology, and especially don't attest to functional similarity.

I think the authors do a good job in arguing for the structure/function divergence, but the entrenched nature of the colicin "link" for many in the field might require more effort to dislodge (so, this is a minor point). It might be useful to emphasize that 4-helix bundles represent a massive family with very divergent functions centered on a common scaffold, and that unless there is something beyond the simple similarity in helical positioning (generally there in all members of the family to define the fold), there is no reason to assume functional similarity. Then the arguments for how function appears to differ in so many ways (as the authors note) might be better contextualized.

However, this might be overkill, but I do think the supplementary figure as is might obscure the divergence for some rather than illuminate it.

We thank the reviewer for comments about Suppl. Fig. 5 and we understand the concerns. It was a bit of a stretch to illustrate any structural superposition of Apo1 NTD with Colicin A or Bcl-xL. We have changed the figure (below). It now permits the reader to assess the structural differences in these proteins in a simplified way with no inference of functional similarity. The previous figure did not illustrate all of the possible superpositions that DALI or PDBeFold would have assessed to provide our conclusions. The figure showed the poor alignment of one example, chosen as was described in the previous caption. The idea was to illustrate, if we chose helix "x" from APOL1 and helix "y" from Bcl-XL as a way to start, look how poor the rest of the helices overlap. Colicin being so much bigger makes choosing a starting point more complicated, but it doesn't matter. If DALI or PDBeFold had suggested a useful match, then that match would have been illustrated.

We thank the reviewer for pointing out the confusion the prior figure may have caused.

Supplementary Figure 5

Supplementary Figure 5. Comparison of the four-helix core region of ApoL1-NTD (helices 1-4: blue color) with colicin-A-PFD (grey color; PDB 1COL) and Bcl-xL (yellow-green color; PDB 4QVE), which serves as an example of the structurally conserved members of the pro-survival protein family. All structures are shown with their α -helices as cylinders. The four-helix core of ApoL1-NTD from the Fab3B6 structure (chain C) was used for the propose of illustration.(center) to show the diverse structural fold between this class of molecules.

REVIEWERS' COMMENTS:

Reviewer #2 (Remarks to the Author):

To my reading the authors have addressed the concerns of all three reviewers adequately. Of course more experimental data would have been great- but I understand the constraints.

Reviewer #3 (Remarks to the Author):

The authors have adequately addressed our comments and we recommend publication.